# A Distance-based Anomaly Detection Framework for Deep Reinforcement Learning

**Hongming Zhang**[*]  *hongmin2@ualberta.ca*
*Department of Computing Science, University of Alberta*
*Alberta Machine Intelligence Institute (Amii), University of Alberta*

**Ke Sun**[*]  *ksun6@ualberta.ca*
*Department of Mathematical and Statistical Sciences, University of Alberta*
*Alberta Machine Intelligence Institute (Amii), University of Alberta*

**Bo Xu**  *boxu@ia.ac.cn*
*Institute of Automation, Chinese Academy of Sciences*

**Linglong Kong**  *lkong@ualberta.ca*
*Department of Mathematical and Statistical Sciences, University of Alberta*
*Alberta Machine Intelligence Institute (Amii), University of Alberta*

**Martin Müller**  *mmueller@ualberta.ca*
*Department of Computing Science, University of Alberta*
*Alberta Machine Intelligence Institute (Amii), University of Alberta*

**Reviewed on OpenReview:** *https: // openreview. net/ forum? id= TNKhDBV6PA*

## Abstract

In deep reinforcement learning (RL) systems, abnormal states pose significant risks by potentially triggering unpredictable behaviors and unsafe actions, thus impeding the deployment of RL systems in real-world scenarios. It is crucial for reliable decision-making systems to have the capability to cast an alert whenever they encounter unfamiliar observations that they are not equipped to handle. In this paper, we propose a novel Mahalanobis distance-based (MD) anomaly detection framework, called *MDX*, for deep RL algorithms. MDX simultaneously addresses random, adversarial, and out-of-distribution (OOD) state outliers in both offline and online settings. It utilizes Mahalanobis distance within class-conditional distributions for each action and operates within a statistical hypothesis testing framework under the Gaussian assumption. We further extend it to robust and distribution-free versions by incorporating Robust MD and conformal inference techniques. Through extensive experiments on classical control environments, Atari games, and autonomous driving scenarios, we demonstrate the effectiveness of our MD-based detection framework. MDX offers a simple, unified, and practical anomaly detection tool for enhancing the safety and reliability of RL systems in real-world applications.

## 1 Introduction

Deep reinforcement learning (RL) algorithms vary considerably in their performance and are highly sensitive to a wide range of factors, including the environment, state observations, and hyper-parameters (Jordan et al., 2020; Patterson et al., 2020). The lack of robustness in RL algorithms and raised safety concerns surrounding learned policies hinder their deployment in real-world scenarios, particularly in safety-critical applications such as autonomous driving (Kiran et al., 2021; Liu et al., 2022; Hu et al., 2023). Recently,

---

[*]These authors contributed equally.

the reliability of RL algorithms has garnered substantial attention (Chan et al., 2020; Gu et al., 2024). Several studies have highlighted the importance of anomaly detection as a crucial component for enabling safe RL systems (García & Fernández, 2015; Hendrycks et al., 2021; Müller et al., 2022), emphasizing the need for anomaly detection-based strategies to build trustworthy and safe RL systems (Sedlmeier et al., 2020a; Danesh & Fern, 2021; Haider et al., 2023).

**Practical Scenarios.** Observed states often contain natural measurement errors (random noises), adversarial perturbations, and out-of-distribution (OOD) observations. For instance, consider an autonomous vehicle with malfunctioning or unreliable sensors or cameras. Under such circumstances, the collected data, such as the vehicle's observed location, can be contaminated by random measurement errors. Furthermore, an autonomous car can encounter sensory inputs that have been adversarially manipulated regarding traffic signs. For example, a stop sign maliciously altered to be misclassified as a speed limit sign (Chen et al., 2019), increases the risk of traffic accidents. Regarding OOD samples, an RL policy trained to drive only on sunny days will struggle with observations from rainy days, which are beyond its trained experience. Such OOD observations can lead to safety violations, performance degradation, and potentially catastrophic failures. All these scenarios highlight the necessity of detecting inaccurate sensor signals from noisy state observations to ensure a vehicle's accurate and reliable operation. Beyond autonomous driving, anomaly detection is critical in many other applications involving sequential decision-making. In healthcare, the RL agent might adjust treatment recommendations if it detects a sudden change in the patient's health condition (Hu et al., 2022). Similarly, detecting fraud and anomalous market states in financial systems is becoming increasingly instrumental in preventing substantial financial losses from market manipulation and fraudulent activities (Hilal et al., 2022).

**Motivating Examples.** Fig. 1(a) illustrates a potential collision scenario where an autonomous car, relying on noisy location data in the red region (such as GPS coordinate errors), turns right prematurely, risking an accident. Without anomaly detection, the car reacts incorrectly due to the location error. Fig. 1(b) highlights how increasing measurement errors, represented by the standard deviation of Gaussian noises, dramatically degrade policy performance. For instance, autonomous cars with RL systems may take sub-optimal or unsafe actions when processing noisy sensory signals in deployment. In addition, incorporating excessive noise during online training (Fig. 1(c)) can severely impair policy learning and diminish performance. These motivating examples underscore the importance of detecting different types of abnormal states for developing trustworthy RL systems in real-world applications.

Our research aims to provide a general framework for applying anomaly detection in deep RL problems, including problem formulation, detection algorithms, and evaluation scenarios. This study contributes to anomaly detection, particularly within the context of safe RL, which falls under the broader research field

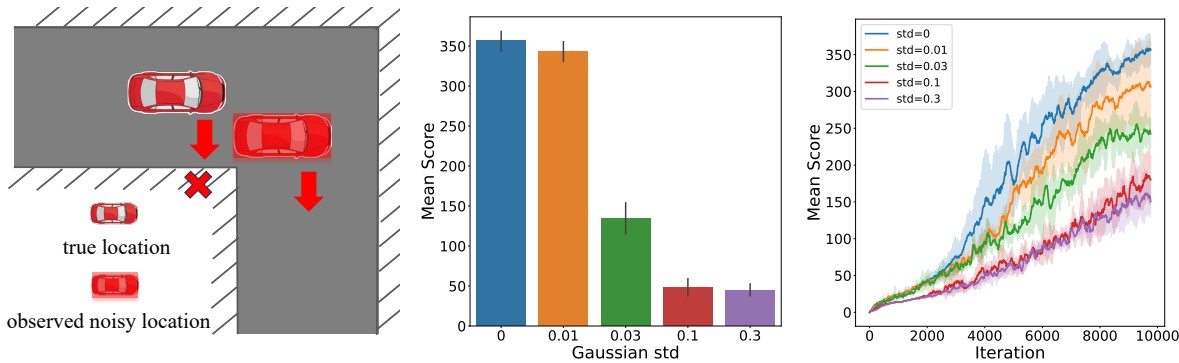

(a) Unsafe behavior in autonomous driving under noisy sensor signals.

(b) Performance degradation when noises injected in policy deployment.

(c) Performance degradation when noises injected during policy learning.

Figure 1: (a) An autonomous car navigates using location data observed from sensors such as GPS. Without an effective anomaly detection mechanism, inaccuracies or malfunctions in these sensors can cause the car to prematurely turn right, leading to a collision. (b) and (c): Performance degradation occurs when noisy states are observed in the Breakout environment. Gaussian noises with increasing standard deviations are injected into the state observations during policy deployment (b) and policy learning (c).

of managing distribution shift in RL; see Section 2 for detailed discussions. Specifically, we strive to develop an effective and unified anomaly detection framework for deep RL in *both offline and online settings*.

1. **Offline Setting.** In this setting, a dataset is fixed without additional online data collection. Given a pre-trained policy, our objective is to utilize a fixed dataset to develop a distance-based anomaly detector tailored for a pre-trained policy. This detector aims to effectively identify whether a state is an outlier [1], ensuring the reliable operation and stable performance of decision-making systems in deployment.

2. **Online Setting.** In this setting, the RL agent interacts with a noisy environment and continuously updates its policy. Our goal is to develop a detection strategy that identifies state outliers, which are outside the RL system's training experience. Removing these outliers can prevent them from interfering with policy training, leading to a robust learning process of RL systems..

Methodologically, we first design an RL outlier detection approach using Mahalanobis Distance (MD) (De Maesschalck et al., 2000) within a statistical hypothesis test framework and extend it to a robust MD version (Butler et al., 1993). These strategies are applied *in a parametric manner* under the Gaussian assumption for state features in each class, which may not always be accurate in practice. To address this limitation, we introduce a *non-parametric conformal version* of MD detection to relax the Gaussian assumption. We empirically investigate the effectiveness of these proposed detection approaches in both offline and online settings across a representative set of RL environments, including classical control environments, Atari games, and autonomous driving. Our contributions can be summarized as follows:

- Our primary technical contribution is the design of RL outlier detection strategies based on the concepts of Mahalanobis Distance (MD), robust MD, and conformal inference. The anomaly detection strategies are specially developed for deep RL within a hypothesis test framework, accommodating both parametric (Gaussian assumption) and non-parametric (conformal calibration) approaches.

- Secondly, in our online setting, our anomaly detection can be applied to a dynamic dataset, where the RL policy continually improves when interacting with the environment. This dynamic setting contrasts with the simpler anomaly detection in supervised learning with a static dataset. To address this challenge, we particularly develop *moving window estimation* and *double self-supervised detectors* for anomaly detection in the online RL setting.

- To our best knowledge, we are the first to conduct a comprehensive study on distance-based anomaly detection in deep RL, covering all typical types of outliers. Our anomaly detectors can simultaneously identify random, adversarial, and out-of-distribution state outliers. We perform extensive experiments to verify the effectiveness of our proposed methods in both offline and online settings.

## 2 Related Work

**Anomaly Detection in Reinforcement Learning.** Anomaly detection has yet to be extensively explored in RL. The connection between anomaly detection and RL was first established in (Müller et al., 2022); however, their work is mainly conceptual and does not propose practical detection algorithms. Change point detection has been investigated in the tabular setting of RL, particularly in environments described as doubly inhomogeneous under temporal non-stationarity and subject heterogeneity (Hu et al., 2022). They focus on identifying "best data chunks" within the environment that exhibit similar dynamics for policy learning, while our detection focuses on anomaly detection in *deep* RL scenarios. Prior studies have also probed anomaly detection in specific RL contexts, such as the offline imitation learning with a transformer-based policy network (Wang et al., 2024) and detecting adversarial attacks within cooperative multi-agent RL (Kazari et al., 2023). However, these studies are limited to specific scenarios that do not address general anomaly detection, even in single-agent RL. Sedlmeier et al. (2020b) introduced a simple policy entropy based

---

[1]Compared with the classical tasks of policy evaluation and learning in offline RL, our offline setting also utilizes a fixed dataset but specifically focuses on developing detection methods given a fixed policy.

out-of-distribution detector in one-class classification problems. Haider et al. (2023) proposed a model-based method using probabilistic dynamics models and bootstrapped ensembles, but this approach highly relies on the capability of the learned environment model and is also computationally expensive. Unlike the previously mentioned detection methods tailored for specific RL areas, our research aims to further enhance this field by developing a distance-based anomaly detection framework applicable to a broad range of deep RL scenarios.

**Distance-based Anomaly Detection.** Recently, there has been a growth of interest in developing anomaly detection strategies in deep learning scenarios (Elmrabit et al., 2020; Pang et al., 2021). In image classification, Mahalanobis distance (MD) was effectively applied by (Lee et al., 2018), who constructed a Mahalanobis confidence score by training a logistic regression detector using validation samples. This score was evaluated in a supervised way, relying on *a validation set*, and thus it is unsuitable for the RL setting. The "tied" covariance assumption used by (Lee et al., 2018), where class-conditional distributions of pre-trained features share the same covariance, was criticized as implausible by (Kamoi & Kobayashi, 2020) based on Gaussian discriminant analysis (Klecka et al., 1980). In contrast, our detection framework MDX avoids the unrealistic "tied covariance" assumption by estimating variance for each class using quadratic discriminant analysis. This approach extends linear boundaries to quadratic ones between classes, offering a more flexible and accurate detection (Hastie et al., 2009). Additionally, we have developed a distribution-free detection strategy using conformal prediction, which eliminates the reliance on the Gaussian assumption and potentially extends applicability across a wider range of practices.

**Robust Statistics for RL.** Deep RL algorithms inherently face challenges related to instability and divergence due to the use of function approximation, bootstrapping, and off-policy learning (Sutton & Barto, 2018). Employing Mahalanobis distance (MD) for anomaly detection can be particularly sensitive during unstable learning phases. The computation of MD is based on Maximum Likelihood Estimate (MLE), which is susceptible to outliers or noisy data (Rousseeuw & Van Zomeren, 1990). Robust statistics (Huber, 2004) have been developed to address these robustness problems, especially leveraging robust estimation techniques that are not unduly affected by outliers. For example, Robust MD is a robust version of MD that employs robust estimators, e.g., Minimum Covariance Determinant (MCD) (Rousseeuw, 1984; Grübel, 1988), for location and covariance estimation (Maronna & Yohai, 2014). Our study enhances the understanding of robust statistical approaches' applicability across a variety of areas in RL, particularly in anomaly detection.

**Conformal Prediction and Conformal Anomaly Detection.** Conformal anomaly detection (Laxhammar & Falkman, 2011; Ishimtsev et al., 2017) is based on the conformal prediction (Angelopoulos et al., 2021; Teng et al., 2023), a popular, modern technique for providing valid prediction intervals for arbitrarily machine learning models. Conformal prediction has garnered increasing attention as it can provide a simple, distribution-free, and computationally effective way of tuning the distribution threshold. Its validity relies on the data exchangeability condition (Shafer & Vovk, 2008), where different orderings of samples are equally likely, but recent studies have verified its applicability in scenarios involving distribution shift (Tibshirani et al., 2019; Barber et al., 2023) and off-policy evaluation (Zhang et al., 2023b). These examples justify the potential of using conformal inference to detect outliers in the context of RL.

**Distribution Shift in RL.** Developing reliable decision-making systems requires effectively addressing distribution shifts in the RL regime. Pertinent research areas include meta RL (Nagabandi et al., 2018; Xu et al., 2018; Ajay et al., 2022), transfer RL (Taylor & Stone, 2009; Parisotto et al., 2015; Zhu et al., 2023; Bai et al., 2024b), continual RL (Khetarpal et al., 2022; Anand & Precup, 2024; Abel et al., 2024), and robust generalization in RL (Boyan & Moore, 1994; Pinto et al., 2017; Zhang et al., 2020). While anomaly detection and these subfields all need to handle distribution shifts to create trustworthy RL systems, our work specifically focuses on detecting outliers to ensure reliable decision-making within the capacity of the learned policy. This focus distinguishes our study from the other related subfields to tackle distribution shifts. For a deeper discussion on their differences, please refer to (Müller et al., 2022).

## 3 Background

**Markov Decision Process.** The interaction of an agent with its environment can be modeled as a Markov Decision Process (MDP), a 5-tuple $(\mathcal{S}, \mathcal{A}, R, P, \gamma)$. $\mathcal{S}$ and $\mathcal{A}$ are the state and action spaces, $P : \mathcal{S} \times \mathcal{A} \times \mathcal{S} \to [0, 1]$ is the environment transition dynamics, $R : \mathcal{S} \times \mathcal{A} \times \mathcal{S} \to \mathbb{R}$ is the reward function and $\gamma \in (0, 1)$ is

the discount factor. The policy $\pi$ is continually updated in this online interaction paradigm. Compared to the online setting, a recent popular paradigm for reinforcement learning is offline RL (Levine et al., 2020). In the offline setting, RL algorithms utilize previously collected data to extract policies without additional online data collection.

**Proximal Policy Optimization (PPO).** The policy gradient algorithm of Proximal Policy Optimization (PPO) (Schulman et al., 2017) has achieved state-of-the-art or competitive performance on Atari games (Bellemare et al., 2013) and MuJoCo robotic tasks (Todorov et al., 2012). Typical policy gradient algorithms optimize the expected reward function $\rho(\theta, s_0) = \mathbb{E}_{\pi_\theta}\left[\sum_{t=0}^{\infty} \gamma^t r(s_t) \mid s_0\right]$ by using the policy gradient theorem (Sutton & Barto, 2018). Here $\pi_\theta$ is the $\theta$-parameterized policy function. Trust Region Policy Optimization (TRPO) (Schulman et al., 2015) and PPO (Schulman et al., 2017) utilize constraints and advantage estimation to perform the update by reformulating the original optimization problem with the surrogate loss $L(\theta)$ as:

$$L(\theta) = \mathbb{E}_t\left[\frac{\pi_\theta(s_t, a_t)}{\pi_{\theta_{\text{old}}}(s_t, a_t)} A_{\pi_{\theta_{\text{old}}}}(s_t, a_t)\right], \tag{1}$$

where $A_{\pi_{\theta_{\text{old}}}}$ is the generalized advantage function (GAE) (Schulman et al., 2018). PPO introduces clipping in the objective function in order to penalize changes to the policy that make $\pi_\theta$ vastly different from $\pi_{\theta_{\text{old}}}$:

$$L^{\text{CLIP}}(\theta) = \mathbb{E}_t\left[\min\left(\frac{\pi_\theta(s_t, a_t)}{\pi_{\theta_{\text{old}}}(s_t, a_t)} A_{\pi_{\theta_{\text{old}}}}(s_t, a_t), \text{clip}\left(\frac{\pi_\theta(s_t, a_t)}{\pi_{\theta_{\text{old}}}(s_t, a_t)}, 1 - \epsilon, 1 + \epsilon\right) A_{\pi_{\theta_{\text{old}}}}(s_t, a_t)\right)\right], \tag{2}$$

where $\epsilon$ is a hyperparameter. We use PPO as the algorithm testbed to examine the efficacy of our anomaly detection framework. However, our detection methods are general and can be easily applied to other RL algorithms (Zhang & Yu, 2020) such as DQN (Mnih et al., 2015; Hessel et al., 2018), A3C (Mnih et al., 2016), and DDPG (Lillicrap et al., 2016; Haarnoja et al., 2018; Fujimoto et al., 2018; Bai et al., 2023).

**Conformal Prediction.** Conformal anomaly detection (Laxhammar & Falkman, 2011; Ishimtsev et al., 2017) is grounded in conformal prediction (Shafer & Vovk, 2008; Angelopoulos et al., 2021), which aims to construct a confidence band $\mathcal{C}_{1-\alpha}(X)$ for $Y$ given a random data pair $(X, Y) \sim \mathcal{P}$ and a confidence level $1 - \alpha$. Suppose we have a pre-trained model $\widehat{\mu}$ and a calibration dataset $(X_1, Y_1), ..., (X_n, Y_n)$ for conformal prediction. We can then compute a predictive interval for the new sample $X_{n+1}$ to cover the unseen response $Y_{n+1}$ by leveraging the empirical quantiles of the residuals $|Y_i - \widehat{\mu}(X_i)|$ on the calibration dataset. This further leads to valid prediction intervals such that:

$$\mathbb{P}(Y_{n+1} \in \mathcal{C}_{1-\alpha}(X_{n+1})) \geq 1 - \alpha, \tag{3}$$

where the confidence band is expected to be as small as possible while maintaining the desired coverage. A fundamental quantity in conformal prediction is the *non-conformity measure*, e.g., the residual $|Y_i - \widehat{\mu}(X_i)|$, which measures how "different" an example is relative to a set of examples (Vovk et al., 2005).

## 4 Mahalanobis Distance-based (MDX) Detection Framework

For a deep RL agent, acting on anomalous inputs could result in hazardous situations. Therefore, developing suitable anomaly detectors for deep RL agents is particularly important in safety-critical scenarios. Fig. 2 illustrates the operational flow of our MDX framework.

**Description of Detection Framework.** Our detection framework is structured around two core components: feature extraction and detector estimation. The process begins by assessing whether a state is anomalous, which is crucially dependent on the associated policy. A state that prompts the policy to initiate a potentially unsafe action is labeled as an outlier. Specifically, we input the state into the policy network and extract the feature vector from the penultimate layer of this network. We categorize states according to the actions determined by the policy, based on the intuition that states associated with the same action share similar features. For each action class, we estimate the mean value ($\mu$) and covariance matrix ($\Sigma$) of the feature vectors as the class centroid. A threshold is set as the class boundary that partitions the feature space into inliers and outliers.

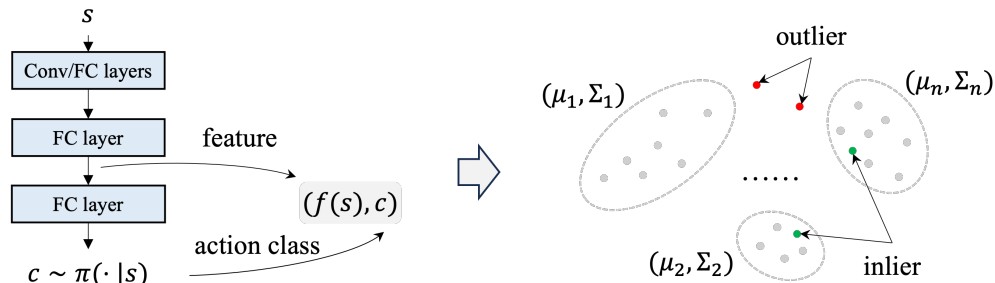

Figure 2: **The detection pipeline of MDX**. We feed the state into the policy network to extract the feature vector and identify its class. For each class, we estimate $(\mu, \Sigma)$ and establish a detection threshold depicted as a dashed ellipse. To determine whether a new state is an outlier, we evaluate its features and compute the distance to the class centroids. If the distance falls below the set threshold, the state is classified as an inlier (green points). Conversely, the state is marked as an outlier (red points).

After the class centroids are estimated, we can determine whether a new state is an outlier by computing its distance from the established class centroids using the Mahalanobis distance based on its feature vector. The Mahalanobis distance is a measure of the distance between a point and a distribution, which in this case is represented by the class centroids. A state is deemed an outlier if the distance surpasses the predefined threshold. An appropriate threshold can balance the trade-off between false positives and false negatives, ensuring that the detection system is both accurate and reliable. By ensuring that only states within the policy's capability are considered valid, MDX enhances the safety and reliability of the RL system.

Our detection framework is generic and can be applied to any agents that operates based on a learned neural network. Based on the proposed detection framework, we instantiate two detection methods in Section 5: distribution-based detection under the assumption of Gaussian distribution, and distribution-free detection, which is based on conformal prediction. The former method relies on Chi-square distribution to determine the threshold, while the latter method employs the conformity score to establish the threshold. We then extend the detection framework to the online setting in Section 6, where the detectors are updated continuously as the agent interacts with the environment.

## 5 Anomaly Detection in the Offline RL Setting

Our MDX detection framework mainly induces two kinds of detection algorithms, including distribution-based detection under Gaussian assumption in Section 5.1 and distribution-free detection by leveraging conformal prediction in Section 5.2. Finally, an integrated detection algorithm pipeline is provided, incorporating all variants of detection strategies in the RL context.

### 5.1 Distribution-based Detection under Gaussian Assumption

**Gaussian Assumption.** The given pre-trained parameterized RL policy $\pi_\theta$ is a discriminative softmax classifier, $\pi(a_t = c|s_t) = \exp\left(\mathbf{w}_c^\top f(s_t) + b_c\right) / \sum_{c'} \exp\left(\mathbf{w}_{c'}^\top f(s_t) + b_{c'}\right)$, where $\mathbf{w}_c$ and $b_c$ are the weight and bias of the policy classifier for action class $c$. The function $f(\cdot)$ represents the output of the penultimate layer of the policy network $\pi_\theta$, serving as the state feature vector. Here, $C = |A|$ is the size of the action space, and $\mu_c$ is the mean vector of $f(s)$ corresponding to the action class $c$ [2]. If we assume that the class-conditional distribution follows a multivariate Gaussian distribution sharing a single covariance $\Sigma$ (tied covariance) in a generative classifier, i.e., $\pi(f(s) \mid a = c) = \mathcal{N}\left(f(s) \mid \mu_c, \Sigma\right)$, then the posterior distribution of $f(s)$ matches the form of a discriminative softmax classifier (Lee et al., 2018). This equivalence implies that $f(s)$ fit a Gaussian distribution under $\pi_\theta$. We approximate state feature vectors with a class-conditional Gaussian

---

[2]Our MDX detection framework currently focuses on environments with the discrete action spaces. For continuous action spaces, a natural solution is to discretize the actions into several bins and then follow the same detection pipeline, which deserves further validation.

distribution with $\mu_c$ and $\Sigma_c$ *for each action class*, rather than using a single "tied" covariance $\Sigma$ across all action classes (Kamoi & Kobayashi, 2020).

**Vanilla MD-based Detection.** An MD-based detection based on Gaussian assumption can be immediately developed based on the mean vectors $\mu_c$ and the covariance matrix $\Sigma_c$ calculated from $f(s)$ for each action class $c$. We first collect $N_c$ state action pairs $\{(s_i, a_i)\}$, separately for each action class $c$, and compute the empirical class mean and covariance of $c$:

$$\widehat{\mu}_c = \frac{1}{N_c} \sum_{i:a_i=c} f(s_i), \quad \widehat{\Sigma}_c = \frac{1}{N_c} \sum_{i:a_i=c} (f(s_i) - \widehat{\mu}_c)(f(s_i) - \widehat{\mu}_c)^\top. \tag{4}$$

In distance-based detection, a straightforward metric is Euclidean distance (ED). However, MD generally outperforms ED in many tasks (Lee et al., 2018; Kamoi & Kobayashi, 2020; Ren et al., 2021), as it incorporates the additional data covariance information to normalize the distance scales. Following the estimation in Eq. (4), we derive the class-conditional Gaussian distribution to characterize the data structure within the state representation space for each action class. For each state $s$ observed by the agent, we compute its *Detection Mahalanobis Distance $M(s)$* between $s$ and the nearest class-conditional Gaussian distribution by:

$$M(s) = \min_c (f(s) - \widehat{\mu}_c)^\top \widehat{\Sigma}_c^{-1} (f(s) - \widehat{\mu}_c). \tag{5}$$

Unlike the previous work Lee et al. (2018), which defined a Mahalanobis confidence score based on a binary classifier in a validation dataset, we utilize $M(s)$ as the detection metric within a statistical hypothesis test framework. Proposition 1 demonstrates that $M(s)$ follows a Chi-squared distribution under the Gaussian assumption.

**Proposition 1.** *(Test Distribution of Detection Mahalanobis distance $M(s)$) Let $f(\mathbf{s})$ be the p-dimensional state random vector for action class $c$. Under the Gaussian assumption $P(f(\mathbf{s})|a = c) = \mathcal{N}(f(\mathbf{s}) \mid \mu_c, \Sigma_c)$, the Detection Mahalanobis Distance $M(\mathbf{s})$ in Eq. (5) is Chi-Square distributed: $M(\mathbf{s}) \sim \chi_p^2$.*

Please refer to Appendix A for the proof. Based on Proposition 1, we can define a threshold $\Theta = \chi_p^2(1 - \alpha)$ by selecting a $\alpha$ value from the specified Chi-Squared distribution to distinguish normal states from outliers. Given a new state observation $s$ and a confidence level $1 - \alpha$, if $M(s) > \Theta$, $s$ is detected as an outlier.

**Robust MD-based Detection.** The estimation of $\mu_c$ and $\Sigma_c$ in Eq. (4) relies on Maximum Likelihood Estimate (MLE), which is sensitive to the presence of outliers in the dataset (Rousseeuw & Van Zomeren, 1990). As the offline data collected from the environment tends to be noisy, directly introducing MD for outlier detection in RL easily results in a less statistically effective estimation of $\mu_c$ and $\Sigma_c$, thus undermining the detection accuracy for outliers. This vulnerability of the MD-based detector against noisy states prompts us to instantiate MDX with a more robust estimator (Huber, 2004).

To this end, we apply the Minimum Covariance Determinant (MCD) estimator (Hubert & Debruyne, 2010) to estimate $\mu_c$ and $\Sigma_c$ by only using a subset of all collected samples. It only uses the observations where the determinant of the covariance matrix is as small as possible. Concretely, MCD determines the subset $J$ of observations with a size $h$, while minimizing the determinant of the sample covariance matrix calculated

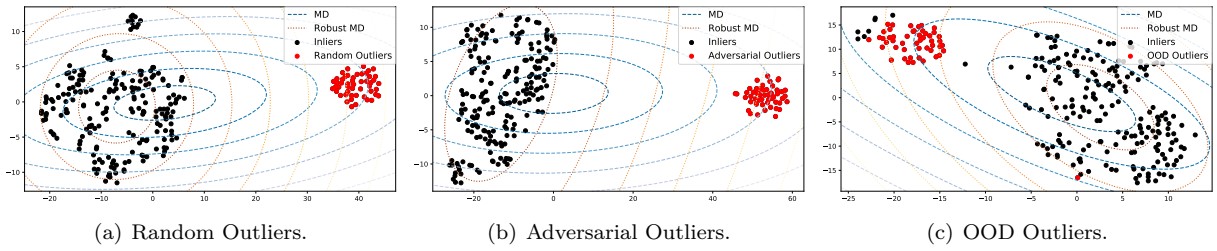

(a) Random Outliers.    (b) Adversarial Outliers.    (c) OOD Outliers.

Figure 3: Contours under the estimation based on MD and Robust MD across different outlier types on Breakout. Black and red points denote inliers and outliers, respectively. The dimension of state feature vectors after a pre-trained PPO policy is reduced by t-SNE (Van der Maaten & Hinton, 2008).

solely from these $h$ points. The choice of $h$ determines the trade-off between the robustness and efficiency of the estimator. The robust MCD mean vector $\widehat{\mu}_c^{\mathrm{rob}}$ and covariance matrix $\widehat{\Sigma}_c^{\mathrm{rob}}$ in the action class $c$ are computed as

$$\widehat{\mu}_c^{\mathrm{rob}} = \frac{1}{h} \sum_{i:i \in J, a_i = c} f(s_i), \quad J = \left\{ \text{ set of } h \text{ points } : \left|\widehat{\Sigma}_J\right| \leq \left|\widehat{\Sigma}_K\right| \text{ for all subsets K} \right\}, \tag{6}$$

where we set $h$ as (`number_of_samples` + `number_of_features` + $1)/2$ (Rousseeuw, 1984). $K$ represents the total number of subsets that contain $h$ points. In practice, the MCD estimator can be efficiently solved by the FAST-MCD algorithm (Hubert & Debruyne, 2010) instead of performing a brute-force search over all possible subsets. Akin to Mahalanobis Distance, we define the *Detection Robust Mahalanobis Distance* $M_{\mathrm{rob}}(s)$ as robust detection metric:

$$M_{\mathrm{rob}}(s) = \min_c \left( f(s) - \widehat{\mu}_c^{\mathrm{rob}} \right)^\top \widehat{\Sigma}_c^{\mathrm{rob}-1} \left( f(s) - \widehat{\mu}_c^{\mathrm{rob}} \right). \tag{7}$$

Since the robust Mahalanobis distance can still approximate the true Chi-squared distribution (Hardin & Rocke, 2005), we still employ the threshold value $\Theta = \chi_p^2(1 - \alpha)$ for detecting outliers as in the MD case.

As a motivating example, Fig. 3 displays contours computed by both MD and Robust MD detection methods for state feature vectors in the Breakout game from the popular Atari benchmark (Bellemare et al., 2013; Brockman et al., 2016) with different types of outliers. These results demonstrate that estimation based on Robust MD is less vulnerable to outlying states (red points) and better fits inliers (black points) than MD. This robust parameter estimation highlights the potential advantage of Robust MD for RL outlier detection, where the data used for estimation tends to be noisy.

## 5.2 Distribution-free Detection by Conformal Inference

Although robust MD-based detection is less vulnerable to noise in RL environments, both MD and robust MD strategies heavily rely on the Gaussian assumption to construct the detection thresholds based on Proposition 1. This distribution assumption is often violated in practice, diminishing the effectiveness of MD and robust MD. In contrast, conformal prediction offers a mathematical framework that provides valid and rigorous prediction distribution without assuming a specific underlying data distribution. The resulting conformal anomaly detection circumvents the limitation of the distribution assumption, potentially improving the detection efficacy.

In the context of RL, conformal anomaly detection evaluates how a state conforms to a model's current prediction distribution, thereby discriminating abnormal states. As a distribution-free detection approach, conformal anomaly detection can enhance the distance-based detectors by additionally tuning the anomaly threshold in the calibration dataset. To design the conformal anomaly detection method, we leverage the Detection Mahalanoibis Distance $M(s)$ as the *non-conformity score*, which measures how dissimilar a state is from the instances in the calibration set. Following split conformal inference (Papadopoulos et al., 2002; Shafer & Vovk, 2008), we split the the previously collected offline dataset into the the calibration set $\mathcal{D}_{\mathrm{cal}}$ and the evaluation set. A simple way is to evaluate the quantiles of the resulting empirical distribution to create the corresponding confidence band. Using the calibration set $\mathcal{D}_{\mathrm{cal}}$, we define the fitted quantiles $\widehat{Q}_{1-\alpha}^c$ of the conformity scores for the action class $c$ as follows:

$$\widehat{Q}_{1-\alpha}^c = \inf \left\{ q : \left( \frac{1}{N_c} \sum_{s_i \in \mathcal{D}_{\mathrm{cal}}, a_i = c} \mathbf{1}_{\{M^c(s_i) \leq q\}} \right) \geq 1 - \alpha \right\}, \tag{8}$$

where each $(s_i, a_i)$ is drawn from the calibration set $\mathcal{D}_{\mathrm{cal}}$ and $c$ is calculated by $c = \arg\min M^c(s_i)$ in $M^c(s_i)$ among all action classes. Finally, we use the class-dependent and well-calibrated detection thresholding $\Theta = \widehat{Q}_{1-\alpha}^c$ in conformal MD-based detection instead of $\chi_p^2(1 - \alpha)$ used in MD and Robust MD strategies.

## 5.3 An Integrated MD-based Detection Algorithm in the Offline Setting

---

**Algorithm 1** MDX Detection Framework in the Offline Setting

---

1: **Input**: The given policy $\pi_\theta$, the dimension of state feature vectors $p$, and a confidence level $1 - \alpha$.
2: **Output**: Detection labels $\{y_s\}$ for each $s$ in the evaluation trajectory.
3: / * Step 1: Detection Design by Estimating Mean and Covariance * /
4: Given state action pairs $\{(s_i, a_i)\}$ where $a_i \sim \pi_\theta(\cdot|s_i)$.
5: **for** each action class $c$ **do**
6:     **if** we choose MD detection **then**
7:         Estimate $\widehat{\mu}_c$ and $\widehat{\Sigma}_c$ via Eq. (4). / * Approach 1: MD Detection * /
8:     **else if** we choose Robust MD detection **then**
9:         Estimate $\widehat{\mu}_c^{\mathrm{rob}}$ and $\widehat{\Sigma}_c^{\mathrm{rob}}$ via Eqs. (6) and (7). / * Approach 2: Robust MD Detection * /
10:     **else**
11:         Estimate $\widehat{\mu}_c, \widehat{\Sigma}_c$ via Eq. (4), calibrate $\widehat{Q}_{1-\alpha}^c$ via Eq. (8) / * Approach 3: Conformal MD-based Detection * /
12:     **end if**
13: **end for**
14: / * Step 2: Detection Deployment * /
15: **for** $s$ in the noisy environment **do**
16:     Compute distance $d = M(s)$ or $d = M_{\mathrm{rob}}(s)$, and threshold $\Theta = \chi_p^2(1 - \alpha)$ or $\Theta = \widehat{Q}_{1-\alpha}^c$.
17:     Set Detection label $y_s = 1$ if $d > \Theta$ else $y_s = -1$.
18: **end for**

---

In practice, it is unclear whether the collected data is noisy or to what extent the Gaussian assumption is violated. Therefore, we provide an integrated algorithm pipeline. Algorithm 1 summarizes all the variant detection strategies of MDX in the offline setting. We compute the (robust) mean vector and covariance matrix among the state feature vectors in the penultimate layer of $\pi_\theta$ for each action class. Next, given a state observation $s$, we compute the detection Mahalanobis distance $d = M(s)$ or $d = M_{\mathrm{rob}}(s)$ and compare it with the threshold $\Theta = \chi_p^2(1 - \alpha)$ under the Gaussian assumption or $\Theta = \widehat{Q}_{1-\alpha}^c$ from distribution-free conformal quantiles. If $d > \Theta$, $s$ is detected as an outlier. Conversely, if $d \leq \Theta$, $s$ is identified as an inlier.

## 6 Anomaly Detection in the Online RL Setting

In the online RL setting (Sutton & Barto, 2018; Dong et al., 2020), a policy is updated continuously, unlike the fixed pre-trained policy used in our offline setting. Robust policy training with noisy states is crucial in safe RL, as the agents are more likely to encounter state outliers during training. In this section, we extend MDX to the online RL training scenario. Unlike the offline setting, the challenge here stems from the dynamic nature of policy updates, requiring our detector to adapt to the evolving distribution of feature vector outputs. The complexity increases when the improved policy starts gathering new samples through exploration, posing a fundamental challenge in an online RL framework. An effective detection system must differentiate between actual noisy observations and newly collected data through exploration. Training the RL agent and estimating the detector are interleaved in a noisy online environment. Various options for managing detected outliers during training include removing or denoising the outlier states. In our detection framework, we focus on direct removal and assess the resulting learning curves in the presence of noisy states during the training process. To address the challenges in detecting abnormal states in the online training setting, we propose *Moving Window Estimation* and *Double Self-supervised Detectors*, both of which are pivotal for the empirical success of our anomaly detection approach.

**Moving Window Estimation.** In the online setting, improving the policy $\pi_\theta$ causes a shift in the data distribution within the replay buffer as the agent interacts with the environment (Rolnick et al., 2019; Xiao et al., 2019). To effectively utilize information from the updated data distribution, we maintain a moving window to store experiences throughout the interaction steps. The moving window operates like a first-in-first-out buffer, storing the most recent samples and discarding the oldest ones. Data falling outside the window is cast away. The moving window can be adjusted to either prioritize a long historical context with a larger window size or consider only more recent experiences with a smaller size. In our experiments, since

---

**Algorithm 2** MDX Detection Framework in the Online Setting, PPO Style

---

1: Initialize policy network $\pi_\theta$ and estimator $\widehat{\mu}_c$ and $\widehat{\Sigma}_c$.
2: Initialize confidence level $1 - \alpha$, the moving window size $m$, inlier and outlier buffers $\mathcal{B}_I$, $\mathcal{B}_O$.
3: **for** iteration $= 1, 2, ..., K$ **do**
4:    **for** actor $= 1, 2, ..., N$ **do**
5:       Run policy $\pi_\theta$ in environment for $T$ timesteps.
6:       Compute distance $d = M(s)$, and threshold $\Theta = \chi_p^2(1 - \alpha)$ or $\Theta = \widehat{Q}_{1-\alpha}^c$.
7:       **if** $d \leq \Theta$ **then**
8:          Add it to $\mathcal{B}_I$.
9:       **else**
10:          Add it to $\mathcal{B}_O$.
11:       **end if**
12:    **end for**
13:    Optimize policy $\pi_\theta$ using inlier trajectories.
14:    Update $\widehat{\mu}_c$ and $\widehat{\Sigma}_c$ based on data in the moving window every $N_c$ new samples come.
15: **end for**

---

the environments we consider have finite horizons with restarts, catastrophic forgetting is not a concern. We set a small window size of 5120 to balance past and recent data used for detection estimation. Based on the constantly updated state feature vectors, $\mu_c$ and $\Sigma_c$ are continually estimated. This continuous updating allows us to accurately track the state feature distribution, ensuring that our detector remains sensitive to recent and historical data shifts.

**Double Self-Supervised Detectors.** Our current detector is continually refined using self-detected inliers, while any detected outliers are promptly discarded. However, a more practical approach is to leverage these outliers to create a complementary detector for outliers. This secondary self-supervised detector validates the detection results from the primary detector. For example, if the primary detector classifies a state as an inlier and the secondary detector agrees that it is not an outlier, the state is confidently classified as such. Conversely, if there is a difference between the discrimination of the two detectors, the state is randomly classified as either an outlier or an inlier. In the event of disagreement, this random classification is motivated by the need to avoid systematic bias that could arise from consistently favoring one detector's output over the other. By introducing randomness, we ensure the system remains fair and does not overly rely on potentially flawed outputs from either detector. This approach also preserves the system's ability to learn and adapt over time, preventing the reinforcement of incorrect classifications. The double-detector system thus enhances the robustness and reliability of the detection process, ensuring more accurate and consistent identification of abnormal states.

**MD-based Detection Algorithm in the Online Setting.** Algorithm 2 outlines our MD-based detection procedure for online RL, incorporating both moving window estimation and double self-supervised detectors. To update our double detectors, inliers and outliers are stored in buffers $\mathcal{B}_I$ and $\mathcal{B}_O$, respectively. For each class, a window size $m$ is specified. Within each class, the state-action pairs in the window are used to estimate $\widehat{\mu}_c$ and $\widehat{\Sigma}_c$. These parameters are updated after every $N_c$ newly collected data points in the window for action class $c$. This adaptive updating mechanism ensures that the detectors remain responsive to evolving data distributions.

**Online Anomaly Detection Procedure.** Since our detection method relies on features extracted from the penultimate layer of the policy network, instead of training the policy from scratch, we pretrain the policy to ensure these features capture meaningful information about the environment. This pre-trained policy results in more meaningful state features and enhances the detection procedure, contributing to a rapid assessment among distinct detection algorithms through their learning curves. Moreover, deploying a randomly initialized policy in a real-world scenario is unreliable. Instead, it is common practice to use a pre-trained policy as a warm start and then further improve it. For example, in recommendation systems, a pre-trained policy is deployed initially to provide recommendations, and user feedback, such as click-through rates (CTR), is used to iteratively update the online policy. Similarly, within our online detection algorithm,

we pre-train a policy using inlier data as a warm start. After pre-training, the policy is introduced to the noisy environment for further online learning. Throughout this process, our MDX framework is used to identify outliers in the subsequent training phases. We then evaluate the training performance of algorithms equipped with these detection mechanisms. This systematic approach facilitates the gradual refinement of the policy while concurrently integrating outlier detection to enhance robustness in real-world settings.

# 7 Experiments

We first conduct experiments on both feature-input and image-input tasks to verify the effectiveness of our MDX framework in both offline and online settings. For feature-input tasks, we choose two classical control environments in OpenAI gym (Brockman et al., 2016), including Mountain Car (Barto et al., 1983) and Cart Pole (Moore, 1990). For image-input tasks, we choose six Atari games (Bellemare et al., 2013). We divide the six Atari games into two different groups. The first group includes Breakout, Asterix, and SpaceInvaders, which feature nearly static backgrounds. Enduro, FishingDerby, and Tutankham in the second group have time-changing or dramatically different backgrounds, presenting more challenging scenarios. We further conduct experiments on autonomous driving environments (Dosovitskiy et al., 2017) as one potential application. We select Proximal Policy Optimization (PPO) (Schulman et al., 2017) as our baseline RL algorithm. For feature-input classical control tasks, we use a policy network with two fully connected layers, each containing 128 units with ReLU activation functions. For image-input tasks, we use the same network architecture as described in the PPO paper (Schulman et al., 2017).

**Three Types of Outliers. (1) Random Outliers.** We generate random outliers by adding Gaussian noise with zero mean and different standard deviations on state observations, simulating natural measurement errors. **(2) Adversarial Outliers.** We perform white-box adversarial perturbations (Szegedy et al., 2013; Goodfellow et al., 2014b; Cao et al., 2020) on state observations for the current policy, following the strategy proposed in (Huang et al., 2017; Pattanaik et al., 2017). Particularly, we denote $a_w^t$ as the "worst" action, with the lowest probability from the current policy $\pi_t(a|s)$. The optimal adversarial perturbation $\eta_t$, constrained in an $\epsilon$-ball, can be derived by minimizing the objective function $J$: $\min_\eta J(s_t + \eta, \pi_t) = -\sum_{i=1}^n p_i^t \log \pi_t(a_i|s_t + \eta), s.t. \|\eta\| \leq \epsilon$, where $p_w^t = 1$ and $p_i^t = 0$ for $i \neq w$. We solve this minimization problem with the Fast Gradient Sign Method (FGSM) (Goodfellow et al., 2014b), a typical adversarial attack method in the deep learning literature. The resulting adversarial outliers $s_t + \eta_t^*$ force the policy to choose $a_w^t$. **(3) Out-of-Distribution (OOD) outliers.** OOD outliers arise from the disparity in data distribution across different environments. To simulate them, we randomly select states from other environments and introduce them to the current environment. In our experiments, we select images from other Atari games to serve as Out-of-Distribution (OOD) outliers within the considered environment. In the autonomous driving scenario, we designate rainy and nighttime observations as OOD outliers for the primary daytime setting on a sunny day. This deliberate selection of diverse outlier examples enables comprehensive testing of our method's robustness across varied environments.

**Baseline Methods.** A fundamental obstacle in assessing the anomaly detection strategies in RL lies in the scarcity of suitable baselines in deep RL settings as introduced in Section 2. To rigorously substantiate the effectiveness of MDX, we initiate our evaluation by comparing them with the foundational baselines we have developed ourselves and implement two non-distance-based methods. (1) **Euclidean distance (ED)** assumes that all features are independent under the Gaussian assumption with one standard deviation, which can be considered as a simplified version of our MD method with an identity covariance matrix. (2) **MD with Tied covariance (TMD)** follows the tied covariance assumption in (Lee et al., 2018), where features among all action classes share a single covariance matrix estimation. (3) **PEOC** is a policy entropy-based detection method proposed in Sedlmeier et al. (2020b). The authors assume that a successful training process reduces entropy for states encountered during training, which can then be used as a classification score to detect OOD states. (4) **EnvModel** follows the model-based detection algorithm utilizing learned dynamics models and bootstrapped ensembles (Haider et al., 2023). We train five autoencoders as environment transition models in each environment. For the offline setting, the autoencoders are trained based on the dataset, while for the online setting, they are continuously updated. Each autoencoder predicts the next state given the current state and action, and the minimum prediction error among the five models serves as the anomaly detection signal. (5) **MD** is our first proposed method with class-conditional Gaussian assumption. (6)

| Detection Accuracy (%) | Outliers | ED | TMD | PEOC | EnvModel | MD | RMD | MD+C |
|---|---|---|---|---|---|---|---|---|
| Cartpole | Random | 68.0 | 93.9 | 50.0 | 50.0 | 95.4 | 78.9 | **96.2** |
| | Adversarial | 51.1 | 93.2 | 50.0 | 50.0 | 94.5 | 78.7 | **94.8** |
| | OOD | 87.3 | 94.3 | 50.0 | 93.8 | 96.5 | 79.1 | **97.5** |
| MountainCar | Random | 89.5 | 86.4 | 50.0 | 49.9 | 90.6 | 78.1 | **93.6** |
| | Adversarial | 64.0 | 81.0 | 50.0 | 49.7 | 85.4 | 74.3 | **87.1** |
| | OOD | 90.9 | 86.5 | 50.0 | 48.7 | 90.5 | 77.2 | **91.7** |
| **Average** | Random | 78.8 | 90.1 | 50.0 | 50.0 | 93.0 | 78.5 | **94.9** |
| | Adversarial | 57.6 | 87.1 | 50.0 | 49.8 | 89.9 | 76.5 | **90.9** |
| | OOD | 89.1 | 90.4 | 50.0 | 71.3 | 93.5 | 78.2 | **94.6** |
| | **Average** | 75.1 | 89.2 | 50.0 | 57.0 | 92.1 | 77.7 | **93.5** |

Table 1: **Average detection accuracy** of MD, RMD, and MD+C compared with baselines across different outlier types in two feature-input classical control environments in the **offline** setting. The averages are computed across environments and outlier types. Accuracy is determined by applying detection techniques to the balanced data composed equally of clean and noisy states.

**Robust MD (RMD)** is the robust variant of MD under the Gaussian assumption. (7) **MD+C** uses well-calibrated conformality scores to construct a valid empirical distance distribution instead of relying on the Chi-Squared distribution established upon the Gaussian assumption.

## 7.1 Anomaly Detection in the Offline Setting

In the offline setting, we randomly split the states from the given dataset into calibration and evaluation sets, each containing 50% of the data. The calibration set is used to construct our detectors, and the evaluation set is for testing. We first use PCA to reduce the state feature vectors into a 50-dimensional space. We then apply (robust) MD to estimate mean vectors and covariances and calibrate the conformality score based on

| Detection Accuracy (%) | Outliers | ED | TMD | PEOC | EnvModel | MD | RMD | MD+C |
|---|---|---|---|---|---|---|---|---|
| Breakout | Random | 53.2 | 59.1 | 50.0 | 50.0 | 61.4 | **71.2** | 62.8 |
| | Adversarial | 84.3 | 89.1 | 50.0 | 50.0 | 90.8 | 80.2 | **91.7** |
| | OOD | 56.9 | 47.8 | 50.0 | **97.5** | 49.6 | **79.5** | 50.8 |
| Asterix | Random | 43.9 | 45.1 | 50.0 | 50.0 | 60.3 | **69.5** | 54.7 |
| | Adversarial | 83.7 | 85.6 | 50.0 | 50.0 | 91.7 | 75.2 | **94.0** |
| | OOD | 39.6 | 40.8 | 50.0 | 53.8 | 46.1 | **57.7** | 49.7 |
| SpaceInvader | Random | 51.4 | 63.9 | 50.0 | 50.0 | 70.2 | **79.4** | 68.6 |
| | Adversarial | 70.9 | 90.3 | 50.0 | 50.0 | 96.1 | 81.1 | **96.5** |
| | OOD | 45.3 | 45.9 | 50.0 | 48.8 | 57.0 | **81.0** | 53.6 |
| Enduro | Random | 49.0 | 59.0 | 50.0 | 50.0 | 72.7 | **82.4** | 70.2 |
| | Adversarial | 92.9 | 91.3 | 50.0 | 50.0 | 96.2 | 83.3 | **97.5** |
| | OOD | 57.1 | 74.9 | 50.0 | 47.6 | 80.0 | **83.4** | 63.3 |
| FishingDerby | Random | 48.9 | 66.4 | 50.0 | 50.0 | 69.8 | **85.7** | 66.5 |
| | Adversarial | 86.3 | 92.5 | 50.0 | 50.0 | **97.4** | 87.2 | **97.4** |
| | OOD | 51.3 | 56.2 | 50.0 | 61.7 | 59.1 | **81.5** | 58.5 |
| Tutankham | Random | 50.0 | 47.5 | 50.0 | 50.0 | 49.1 | **74.0** | 49.8 |
| | Adversarial | 66.2 | 89.5 | 50.0 | 50.0 | 95.3 | 77.1 | **96.5** |
| | OOD | 55.0 | 83.3 | 50.0 | **97.5** | 89.6 | 77.2 | 79.7 |
| **Average** | Random | 49.4 | 56.8 | 50.0 | 50.0 | 63.9 | **77.0** | 61.7 |
| | Adversarial | 80.7 | 89.7 | 50.0 | 50.0 | 94.6 | 80.7 | **95.6** |
| | OOD | 50.8 | 58.1 | 50.0 | 67.8 | 63.6 | **76.7** | 59.3 |
| | **Average** | 60.3 | 68.2 | 50.0 | 55.7 | 74.0 | **78.1** | 72.2 |

Table 2: **Average detection accuracy** of MD, RMD, and MD+C compared with baselines across different outlier types in six Atari games in the **offline** setting. The averages are computed across environments and outlier types. Accuracy is determined by applying detection techniques to the balanced data composed equally of clean and noisy states.

the calibration dataset. Finally, we incorporate the three types of noises into the originally clean evaluation dataset. We assess the performance of our detection methods on the entire evaluation dataset.

**Main results.** Tables 1 and 2 show the detection accuracy of MDX instantiated with vanilla MD, robust MD, and conformal MD with $\alpha = 0.05$ across a wide range of outlier types on each task. A higher accuracy indicates a more successful identification of anomalies for the evaluated detection method. We conclude that: (1) All MD-based methods, i.e., TMD, MD, RMD, and MD+C, outperform ED, confirming the usefulness of covariance matrix information in RL outlier detection. (2) MD+C performs consistently best on classic control tasks and excels in identifying adversarial outliers on Atari games. Robust MD generally performs the best on Atari games, significantly surpassing MD and other methods in detecting random and OOD outliers. Nonetheless, robust MD is not effective enough to detect adversarial outliers. We hypothesize that the robustness advantage resulting from RMD in detector estimation is more applicable in image input or the high-dimensional state space. (3) PEOC is almost ineffective across all considered state outliers, suggesting that the entropy difference is useless for identifying outliers. By contrast, EnvModel is only superior to the other approaches against some OOD outliers, which is not generally preferable. More detailed results are provided in Tables 5 and 6 of Appendix B.1.

**Sensitivity Analysis on Feature Dimension Reduction.** We provide a sensitivity analysis on Atari games regarding the number of feature dimensions reduced by PCA, showing that the detection accuracy for all considered outliers tends to improve as the number of principal components increases. This indicates that better detection performance can be achieved with higher feature dimensions. The detailed results are presented in Appendix B.4.

**Effectiveness of Robust MD.** In robust MD analysis, it is typically concluded that outlier states are more distinctly separated from inlier states. By comparing the Mahalanobis distance distributions between inliers and outliers under both MD and Robust MD, we show that this conclusion also applies to the RL anomaly detection scenario. This effect explains the detection advantage of robust MD in RL. Detailed results are provided in Appendix B.3.

## 7.2 Anomaly Detection in the Online Setting

The PPO agent, utilizing multi-processes as detailed in the original PPO algorithm (Schulman et al., 2017), runs eight independent environments in parallel, and we introduce state outliers into four of these environments. For random and adversarial outliers, actions are determined based on the PPO policy network

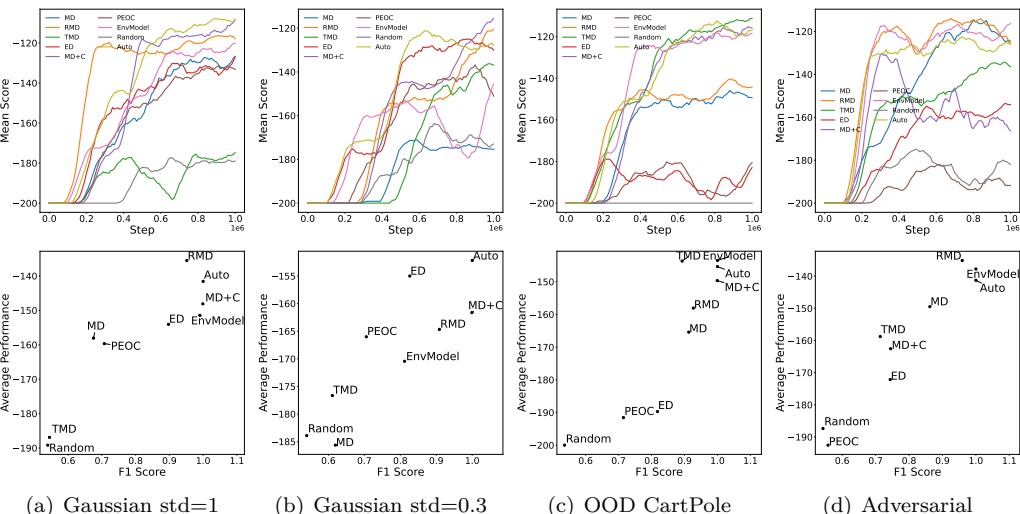

(a) Gaussian std=1     (b) Gaussian std=0.3     (c) OOD CartPole     (d) Adversarial

Figure 4: Performance on MountainCar across various state outliers in online learning. The first row shows the policy performance during learning. The second row shows the relationship between the averaged detection accuracy and performance.

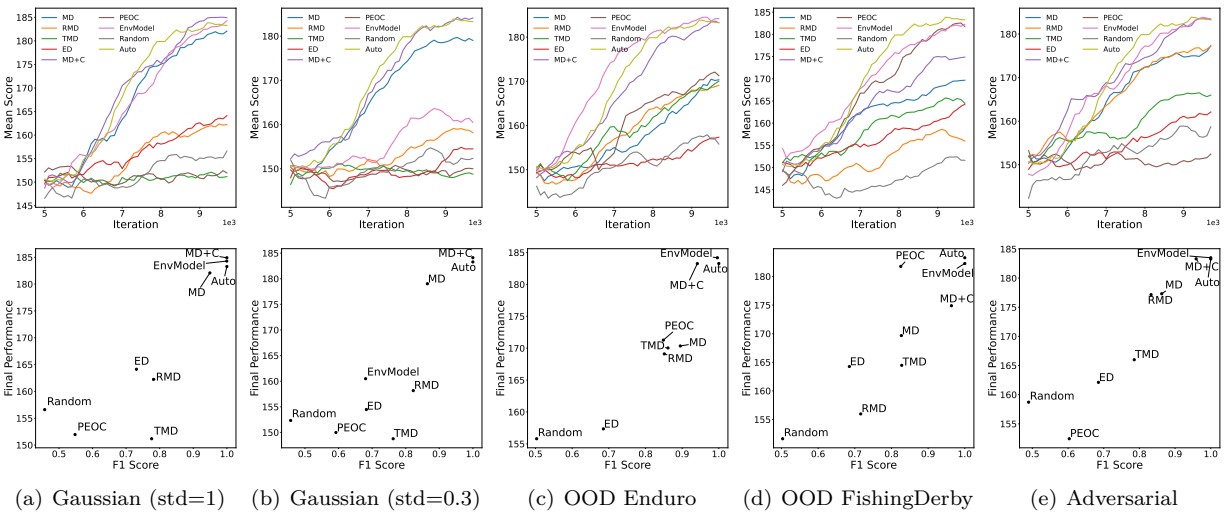

Figure 5: Performance on Tutankham across various state outliers in online learning. The first row shows the policy performance during learning. The second row shows the relationship between the averaged detection accuracy and achieved final performance.

$\pi_\theta$. For OOD outliers, due to the potential differences in action spaces between the original environment and the OOD environment, we select OOD states from the OOD environment by taking random actions within its own action space. For the Robust MD method, we use PCA to reduce state feature vectors into a 50-dimensional space due to the expensive computation of the robust MD method. For the other methods, we use the original feature vectors output from the penultimate layer of $\pi_\theta$. Results are averaged over three seeds with hyperparameters given in Table 7 of Appendix C.1. When our detectors identify an outlier, it is removed from training. We compare the resulting learning curves for different detection methods.

**Additional Baselines.** We add another two baselines as performance upper bound and lower bound. (1) For an ideal baseline, the method **Auto** automatically deletes true state outliers, showing the optimal training performance of algorithms *without the interruption from outliers*. (2) At the other extreme, **Random** uses a totally random detector that detects a state as an inlier or outlier with a probability of 0.5.

**Main Results.** Figs. 4 and 5 present the online performance on feature-input task MountainCar and image-input Atari game Tutankham. The first row shows the learning curves of cumulative rewards based on the PPO algorithm. To better highlight their differences, we omit the confidence bands in Fig. 5, while providing full results with confidence bands in Appendix C.1 Figs. 14 to 21 for reference. The second row illustrates the relationship between detection accuracy and policy performance. For Atari games, the x-axis represents the average F1 score during learning, while the y-axis represents the final performance. For feature-input tasks like MountainCar and CartPole, some methods can achieve the maximum score, leading to no significant difference in final performance. We exhibit the relationship between the average F1 score and the average

| Superiority Rank | Outlier Type | Random | ED | TMD | PECO | EnvModel | MD | RMD | MD+C |
|---|---|---|---|---|---|---|---|---|---|
| | Random | 6.75 | 3.75 | 7.25 | 6.00 | 3.25 | 4.50 | 3.25 | 1.25 |
| Performance | OOD | 7.75 | 6.25 | 4.25 | 5.00 | 1.50 | 3.75 | 4.50 | 3.00 |
| | Adversarial | 7.00 | 6.00 | 5.00 | 8.00 | 1.50 | 3.50 | 2.50 | 2.50 |
| Average | All | 7.20 | 5.20 | 5.60 | 6.00 | **2.20** | 4.00 | 3.60 | **2.20** |
| | Random | 8.00 | 4.75 | 5.75 | 6.00 | 3.00 | 4.25 | 3.25 | 1.00 |
| F1 Score | OOD | 8.00 | 5.75 | 4.75 | 5.75 | 1.75 | 4.25 | 4.00 | 1.75 |
| | Adversarial | 8.00 | 5.50 | 5.50 | 7.00 | 1.00 | 3.00 | 3.00 | 3.00 |
| Average | All | 8.00 | 5.30 | 5.30 | 6.10 | 2.10 | 4.00 | 3.50 | **1.70** |

Table 3: The average superiority rank (1 is best) of anomaly detection methods across all types of outliers. Numbers in bold represent the best results.

policy performance during learning, similar to previous work (Zhang et al., 2023a). We can find that higher detection accuracy is generally associated with better policy performance. For each outlier type in Table 3, we evaluate the **superiority rank** of all detectors regarding the F1 score and policy performance, where rank 1 indicates the best performance. A smaller superiority rank implies a more effective detection. Our conclusions are as follows: (1) Conformal MD (MD+C) generally achieves the best detection performance across all considered baselines (except Auto). The superiority of MD+C over MD highlights the crucial role of accurately calibrated thresholds in the online RL detection setting. (2) PECO is ineffective in most of our experiments, which indicates that policy entropy is not a reliable indicator for detecting anomalies. (3) While EnvModel achieves competitive performance in most tasks, it is less practical because it requires executing the action in the environment to obtain the next state before detecting the current state. That being said, EnvModel is a hindsight method and is less applicable in real applications.

**Ablation Study on Double Self-Supervised Detectors.** We conduct an ablation study of double self-supervised detectors on Breakout with random and OOD outliers. Results in Fig. 22 of Appendix C.2 show that double self-supervised detectors reduce detection errors and improve detection accuracy.

**Ablation Study on Outlier Proportions.** We also demonstrate the robust detection performance across different proportions of outliers encountered by the agent during training. We conduct experiments on Breakout, and the results are provided in Fig. 23 of Appendix C.3.

### 7.3 Autonomous Driving Environment

To verify the broader applicability of our method, we perform experiments on autonomous driving environments based on CARLA (Dosovitskiy et al., 2017) and introduce practical scenarios in which all three types of anomalies commonly occur. Since anomaly detection in autonomous driving is more practical in the offline setting and CARLA is a complex environment that exceeds our computational capacity for online training, we focus on the offline setting in this section.

**Random Noise.** Malfunctioning sensors or cameras can introduce random noise into signal observations. For instance, a faulty camera lens may produce distorted images, while a malfunctioning LiDAR sensor might generate erroneous depth measurements. Such random noise can impair the reliability of perception systems in autonomous vehicles.

**Adversarial Attacks.** Adversarial attacks involve intentionally manipulating input signals to disrupt the functioning of RL systems (Bai et al., 2024a). In the context of autonomous driving, an attacker might tamper with sensor data or traffic signs, resulting in misleading observations and potentially hazardous driving behavior. Adversarial states thus pose a significant threat to the robustness and safety of autonomous driving systems.

**Out-of-Distribution (OOD) States.** Consider a scenario where an RL policy is trained exclusively under sunny weather. Encountering rainy weather poses a challenge, as the observations captured under these conditions deviate from the training data distribution. Such observations are therefore considered Out-of-Distribution (OOD) states.

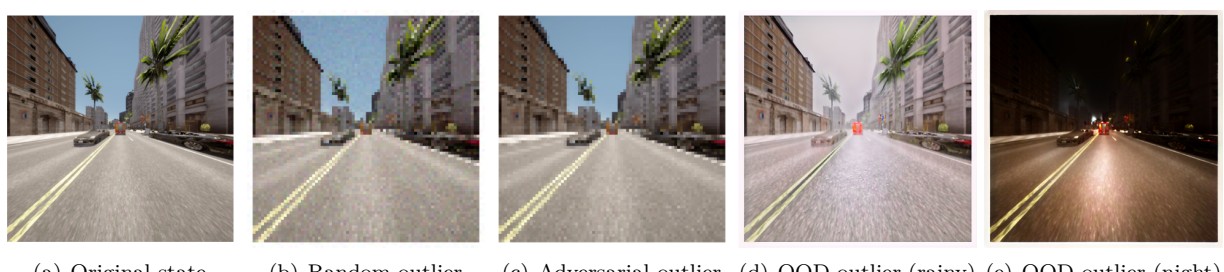

(a) Original state     (b) Random outlier     (c) Adversarial outlier     (d) OOD outlier (rainy)     (e) OOD outlier (night)

Figure 6: The clean and noisy state observations in autonomous driving experiments.

| Detection Accuracy (%) | ED | TMD | PEOC | EnvModel | MD | RMD | MD+C |
|---|---|---|---|---|---|---|---|
| Random (std $\in [0.005, 0.07]$) | 50.0 | 62.0 | 50.0 | 50.0 | 68.7 | **72.1** | 60.7 |
| Random (std $\in (0.07, 0.3]$) | 50.0 | 95.4 | 50.0 | 50.0 | 95.2 | 73.8 | **95.8** |
| Adversarial | 50.0 | 96.4 | 50.0 | 50.0 | 91.0 | 73.8 | **97.5** |
| OOD (Rain) | 50.0 | 96.4 | 50.0 | 52.7 | 95.3 | 73.6 | **97.5** |
| OOD (Night) | 50.0 | 96.4 | 50.0 | **97.5** | 95.3 | 73.8 | **97.5** |

Table 4: Detection accuracy on the CARLA *town* environment over three types of outliers.

**Experimental Setup.** We conduct experiments using the CARLA environment (Dosovitskiy et al., 2017). CARLA is an open-source simulator for autonomous driving research known for its high-quality rendering and realistic physics. The environment includes 3D models of static objects, such as buildings, vegetation, traffic signs, and infrastructure, as well as dynamic objects, such as vehicles and pedestrians. The task is to drive safely through the town. In each episode, the vehicle must reach a given goal without collision. The episode ends when the vehicle reaches the goal, collides with an obstacle, or exceeds the time limit.

**Noisy State Observations.** Following the approach used in Atari game settings, we introduce Gaussian noise to simulate random outliers and generate adversarial outliers using adversarial perturbations. For OOD outliers, we leverage CycleGAN-Turbo (Zhu et al., 2017; Parmar et al., 2024), a technique designed for adapting a single-step diffusion model (Ho et al., 2020) to new tasks and domains through adversarial learning (Goodfellow et al., 2014a). This method can perform various image-to-image translation tasks and outperforms existing GAN-based and diffusion-based methods for various scene translation tasks, such as day-to-night conversion and adding/removing weather effects like fog, snow, and rain (Parmar et al., 2024). Specifically, we use CycleGAN-Turbo to create **rainy** and **nighttime** outliers. Examples of different anomaly states are presented in Fig. 6.

**Main Results.** Given a fixed dataset and a pre-trained policy, we assess our detection methods across the three types of outliers. Table 4 shows the average accuracy, with MD+C achieving the highest performance in most scenarios, while RMD performs best in the presence of small random noises. Similar to the results in classical control environments and Atari games, the entropy-based PEOC is still ineffective across all outlier settings. While EnvModel is competitive in identifying nighttime outliers, it is inferior to the other methods against the other considered outliers. These results suggest that our proposed method effectively detects outliers for realistic problems, such as autonomous driving.

# 8 Discussions and Conclusion

In this paper, we present the first detailed study of a distance-based anomaly detection framework in deep RL, considering random, adversarial, and OOD state outliers in both offline and online settings. The primary detection backbone is based on Mahalanobis distance, and we extend it to robust and distribution-free versions by leveraging robust estimation and conformal prediction techniques. Experiments on classical control environments, Atari games, and the autonomous driving environment, demonstrate the effectiveness of our proposed methods in detecting the three types of outliers. The conformal MD method achieves the best detection performance in most scenarios, especially in the online setting. Our research contributes to developing safe and trustworthy RL systems in real-world applications.

**Limitations and Future Work.** In the online setting, especially with a high proportion of outliers, it may be preferable to denoise the detected state outliers via some neighboring smoothing techniques, e.g., *mixup* (Zhang et al., 2018; Wang et al., 2020), rather than deleting them directly as performed in this paper. To relax the Gaussian assumption in the hypothesis test of our detection, we can consider other non-parametric methods, such as one-class support vector machines (Choi, 2009) or isolation forests (Liu et al., 2008). A substantial challenge that remains for future work is to devise a more informed detector to distinguish between real "bad" outliers that can cause truly misleading actions and "good" *new samples* collected through exploration, which can potentially benefit the policy learning, especially for image inputs (Zhang & Ranganath, 2023).

**Acknowledgements**

Ke Sun was supported by the State Scholarship Fund from China Scholarship Council (No:202006010082). Linglong Kong was partially supported by grants from the Canada CIFAR AI Chairs program, the Alberta Machine Intelligence Institute (AMII), and Natural Sciences and Engineering Council of Canada (NSERC), and the Canada Research Chair program from NSERC. Hongming Zhang and Martin Müller were supported by UAHJIC, the Natural Sciences and Engineering Research Council of Canada (NSERC), and an Amii Canada CIFAR AI Chair.

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

## A    Proof of Proposition 1

*Proof.* We show that for each action class $c$, the square of Mahalanobis distance $d$ is identically independent Chi-squared distributed under the Gaussian assumption. Without loss of generality, we denote $\mu$ and $\Sigma$ as the mean and variance matrix of the closest class-conditional Gaussian distribution. We need to show $d = (f(\mathbf{s}) - \mu)^\top \Sigma^{-1} (f(\mathbf{s}) - \mu)$ is Chi-squared distributed. Firstly, by eigenvalue decomposition, we have

$$\Sigma^{-1} = \sum_{k=1}^{p} \lambda_k^{-1} u_k u_k^\top, \tag{9}$$

where $\lambda_k$ and $u_k$ are the $k$-th eigenvalue and eigenvector of $\Sigma$. Plugging it into the form of $d$, we immediately obtain

$$
\begin{aligned}
d =& (f(\mathbf{s}) - \mu)^\top \Sigma^{-1} (f(\mathbf{s}) - \mu) \\
=& (f(\mathbf{s}) - \mu)^\top (\sum_{k=1}^{p} \lambda_k^{-1} u_k u_k^\top)(f(\mathbf{s}) - \mu) \\
=& \sum_{k=1}^{p} \lambda_k^{-1} (f(\mathbf{s}) - \mu)^\top u_k u_k^\top (f(\mathbf{s}) - \mu) \\
=& \sum_{k=1}^{p} \left[ \lambda_k^{-\frac{1}{2}} u_k^\top (f(\mathbf{s}) - \mu) \right]^2 \\
=& \sum_{k=1}^{p} \mathbf{X}_k^2,
\end{aligned}
\tag{10}
$$

where $\mathbf{X}_k^2$ is a new Gaussian variable that results from the linear transform of a Gaussian distribution $f(\mathbf{s})$ where $f(\mathbf{s}) \sim \mathcal{N}(\mu, \Sigma)$. Therefore, the resulting variance $\sigma_k^2$ can be derived as

$$\sigma_k^2 = \lambda_k^{-\frac{1}{2}} u_k^\top \Sigma \lambda_k^{-\frac{1}{2}} u_k = \lambda_k^{-1} u_k^\top (\sum_{j=1}^{p} \lambda_j u_j u_j^\top) u_k = \sum_{j=1}^{p} \lambda_k^{-1} \lambda_j u_k^\top u_j u_j^\top u_k \tag{11}$$

As the $\mu_j$ and $\mu_k$ are orthogonal if $j \neq k$, the variance $\sigma_k^2$ can be further reduced to

$$\sigma_k^2 = \lambda_k^{-1} \lambda_k u_k^\top u_k u_k^\top u_k = \|u_k\|^2 \|u_k\|^2 = 1. \tag{12}$$

Each $\mathbf{X}_k$ is a standard Gaussian distribution. Then we have $d$, the square of Mahalanobis distance, Chi-squared distributed, i.e., $d \sim \chi^2(p)$, independent of the action class $c$. Without loss of generality, the smallest $d$ among all action classes, i.e., $M(\mathbf{s})$, is also a Chi-squared distribution. That is to say, $M(\mathbf{s}) \sim \chi^2(p)$.

$\square$

## B    Results in Offline Setting

### B.1    Results across Different Noise Strengths

We provide detailed detection accuracy of various detection methods across different noise strengths. The results on two classical control environments and six Atari games are gived in Tables 5 and 6.

### B.2    Visualization of Outlier States on Six Games

We plot the outlier states on Breakout, Asterix, and SpaceInvaders games in Fig. 7 and outliers states on Enduro, FishingDerby, and Tutankham in Fig. 8.

Table 5: Detection accuracy (%) of our MD, Robust MD, and conformal MD strategies compared with other baseline methods on two classical control environments with $\alpha = 0.05$.

| Environments | Outliers | Perturbation | ED | TMD | PEOC | EnvModel | MD | RMD | MD+C |
|---|---|---|---|---|---|---|---|---|---|
| Cartpole | Random | std=0.3 | 60.54 | 93.24 | 50.00 | 50.01 | 94.74 | 78.7 | **95.33** |
| | | std=0.5 | 75.54 | 94.49 | 50.00 | 49.98 | 96.14 | 79.15 | **97.00** |
| | Adversarial | $\epsilon$=0.15 | 50.58 | 92.53 | 50.00 | 49.98 | 93.65 | 78.48 | **93.79** |
| | | $\epsilon$=0.2 | 51.65 | 93.89 | 50.00 | 50.01 | 95.31 | 78.97 | **95.76** |
| | OOD | MountainCar | 87.27 | 94.26 | 50.00 | 93.84 | 96.45 | 79.09 | **97.46** |
| MountainCar | Random | std=0.3 | 86.91 | 85.82 | 50.00 | 50.11 | 89.81 | 77.84 | **92.52** |
| | | std=0.5 | 92.08 | 86.96 | 50.00 | 49.71 | 91.37 | 78.35 | **94.73** |
| | Adversarial | $\epsilon$=0.001 | 63.86 | 80.72 | 50.00 | 49.69 | 85.09 | 74.04 | **86.83** |
| | | $\epsilon$=0.01 | 64.13 | 81.30 | 50.00 | 49.693 | 85.72 | 74.63 | **87.32** |
| | OOD | Cartpole | 90.89 | 86.51 | 50.00 | 48.73 | 90.49 | 77.21 | **91.69** |

Table 6: Detection accuracy (%) of our MD, Robust MD, and conformal MD strategies compared with other baseline methods on six Atari games with $\alpha = 0.05$.

| Games | Outliers | Perturbation | ED | TMD | PEOC | EnvModel | MD | RMD | MD+C |
|---|---|---|---|---|---|---|---|---|---|
| Breakout | Random | std=0.02 | 50.13 | 52.01 | 50.00 | 50.00 | 54.89 | **62.80** | 54.46 |
| | | std=0.04 | 56.18 | 66.26 | 50.00 | 50.00 | 67.85 | **79.64** | 66.76 |
| | Adversarial | $\epsilon$=0.001 | 81.28 | 87.44 | 50.00 | 50.00 | 89.39 | 79.85 | **89.54** |
| | | $\epsilon$=0.01 | 87.36 | 90.67 | 50.00 | 50.00 | 92.23 | 80.57 | **93.87** |
| | OOD | Asterix | 66.80 | 47.91 | 49.99 | **97.50** | 50.32 | 80.73 | 51.15 |
| | | SpaceInvaders | 47.07 | 47.74 | 49.99 | **97.50** | 48.92 | 78.22 | 50.37 |
| Asterix | Random | std=0.1 | 42.56 | 42.87 | 50.01 | 49.99 | 48.28 | **63.05** | 49.04 |
| | | std=0.2 | 45.29 | 47.37 | 50.01 | 49.99 | 72.31 | **75.86** | 60.31 |
| | Adversarial | $\epsilon$=0.001 | 83.41 | 85.31 | 50.01 | 50.00 | 91.38 | 75.02 | **93.62** |
| | | $\epsilon$=0.01 | 83.94 | 85.90 | 50.01 | 50.00 | 91.95 | 75.40 | **94.35** |
| | OOD | Breakout | 41.73 | 42.94 | 50.01 | 48.95 | 48.24 | **75.72** | 51.85 |
| | | SpaceInvaders | 37.38 | 38.61 | 50.01 | **58.55** | 43.92 | 39.62 | 47.50 |
| SpaceInvaders | Random | std=0.02 | 50.00 | 48.46 | 50.01 | 50.00 | 53.26 | **75.24** | 52.53 |
| | | std=0.04 | 52.88 | 79.38 | 50.01 | 50.00 | **87.20** | 83.57 | 84.62 |
| | Adversarial | $\epsilon$=0.001 | 67.13 | 89.69 | 50.01 | 50.00 | 95.64 | 83.16 | **95.87** |
| | | $\epsilon$=0.01 | 74.66 | 90.93 | 50.01 | 50.00 | 96.65 | 79.07 | **97.05** |
| | OOD | Breakout | 45.81 | 45.64 | 50.01 | 48.97 | 56.56 | **78.88** | 54.23 |
| | | Asterix | 44.71 | 46.22 | 50.01 | 48.72 | 57.45 | **83.06** | 53.03 |
| Enduro | Random | std=0.1 | 49.42 | 45.27 | 50.00 | 50.00 | 54.03 | **81.65** | 52.24 |
| | | std=0.2 | 48.67 | 72.70 | 50.00 | 49.99 | **91.40** | 83.15 | 88.17 |
| | Adversarial | $\epsilon$=0.001 | 91.86 | 91.26 | 50.00 | 49.99 | 96.24 | 83.27 | **97.48** |
| | | $\epsilon$=0.01 | 93.93 | 91.26 | 50.00 | 49.99 | 96.24 | 83.37 | **97.48** |
| | OOD | FishingDerby | 63.95 | 83.15 | 50.00 | 47.54 | **85.10** | 83.44 | 61.39 |
| | | Tutankham | 50.19 | 66.60 | 50.00 | 47.67 | 74.98 | **83.28** | 65.13 |
| FishingDerby | Random | std=0.2 | 48.80 | 48.20 | 50.00 | 50.00 | 51.65 | **83.77** | 50.82 |
| | | std=0.3 | 49.03 | 84.60 | 50.00 | 50.00 | **87.86** | 87.71 | 82.09 |
| | Adversarial | $\epsilon$=0.001 | 83.44 | 92.48 | 50.00 | 50.01 | **97.31** | 86.90 | 97.33 |
| | | $\epsilon$=0.01 | 89.12 | 92.54 | 50.00 | 50.01 | **97.49** | 87.41 | 97.46 |
| | OOD | Enduro | 48.89 | 56.69 | 50.00 | 75.34 | 60.31 | **86.45** | 59.78 |
| | | Tutankham | 53.64 | 55.65 | 49.96 | 47.97 | 57.93 | **76.59** | 57.25 |
| Tutankham | Random | std=0.04 | 50.00 | 48.30 | 50.00 | 50.03 | 49.31 | **71.49** | 50.00 |
| | | std=0.06 | 50.00 | 46.68 | 50.03 | 50.00 | 48.79 | **76.51** | 49.57 |
| | Adversarial | $\epsilon$=0.01 | 60.08 | 89.37 | 50.03 | 50.01 | 95.24 | 77.07 | **95.56** |
| | | $\epsilon$=0.05 | 72.36 | 89.55 | 50.03 | 50.01 | 95.28 | 77.05 | **97.49** |
| | OOD | Enduro | 49.96 | 89.17 | 50.03 | **97.50** | 95.12 | 77.18 | 91.59 |
| | | FishingDerby | 50.0 | 77.34 | 50.03 | **97.50** | 84.12 | 77.17 | 67.77 |

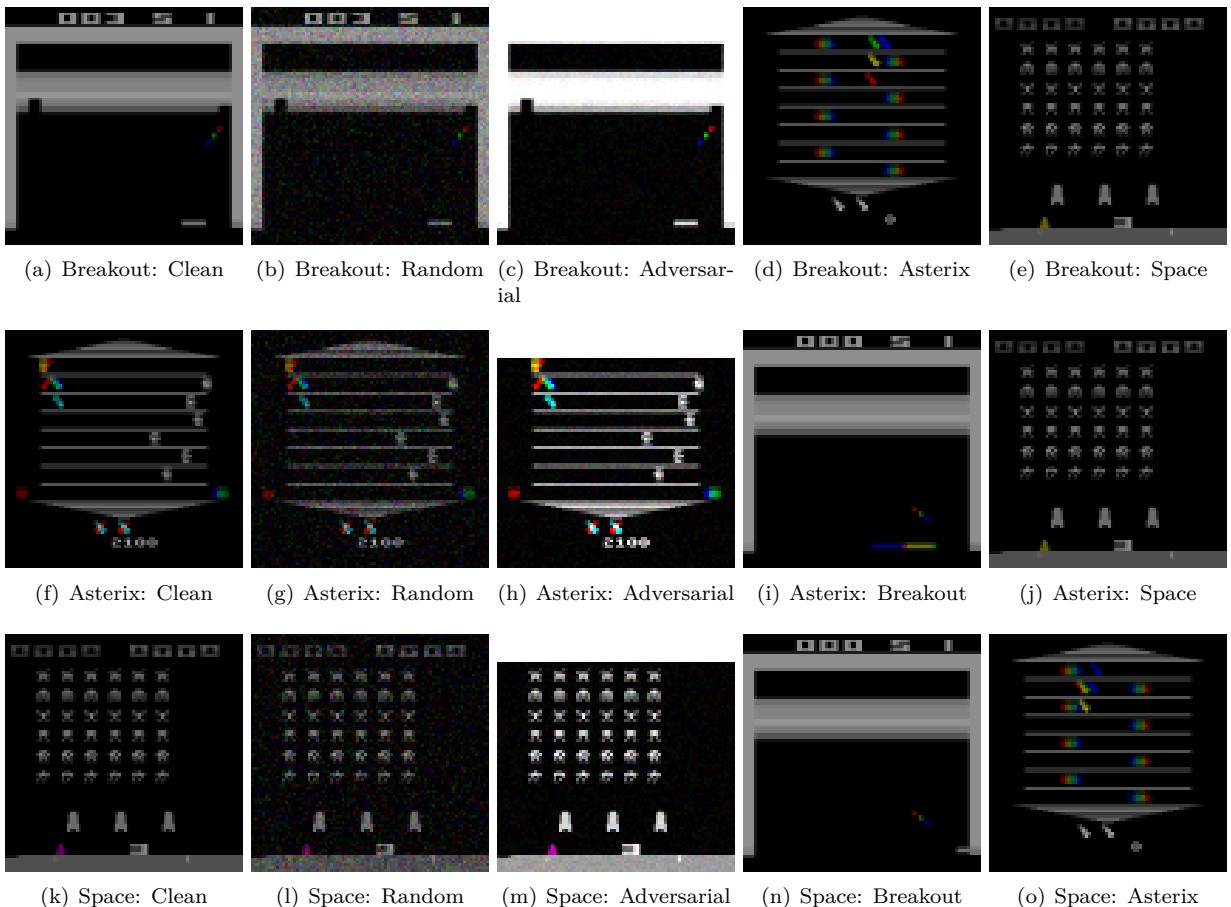

(a) Breakout: Clean     (b) Breakout: Random     (c) Breakout: Adversarial     (d) Breakout: Asterix     (e) Breakout: Space

(f) Asterix: Clean     (g) Asterix: Random     (h) Asterix: Adversarial     (i) Asterix: Breakout     (j) Asterix: Space

(k) Space: Clean     (l) Space: Random     (m) Space: Adversarial     (n) Space: Breakout     (o) Space: Asterix

Figure 7: Visualization of various state outliers on Breakout, Asterix, and SpaceInvaders games.

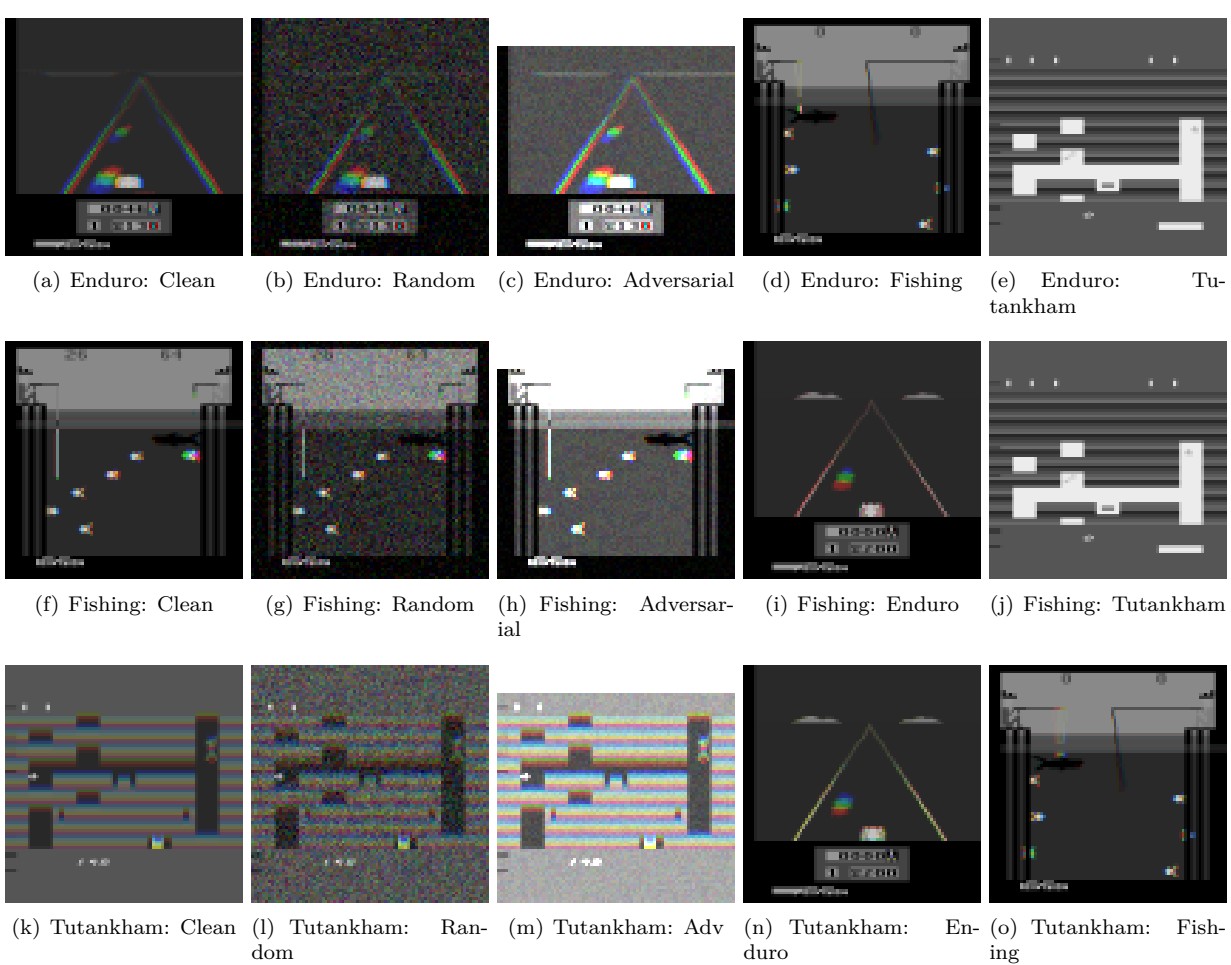

(a) Enduro: Clean    (b) Enduro: Random    (c) Enduro: Adversarial    (d) Enduro: Fishing    (e) Enduro: Tutankham

(f) Fishing: Clean    (g) Fishing: Random    (h) Fishing: Adversarial    (i) Fishing: Enduro    (j) Fishing: Tutankham

(k) Tutankham: Clean    (l) Tutankham: Random    (m) Tutankham: Adv    (n) Tutankham: Enduro    (o) Tutankham: Fishing

Figure 8: Visualization of various state outliers on Enduro, FishingDerby, and Tutankham games.

## B.3 Effectiveness of Robust MD

We take the cubic root of the Mahalanobis distances, yielding approximately normal distributions (Wilson & Hilferty, 1931). In this experiment, 250 clean states are drawn from the replay buffer, and 50 abnormal states are drawn from each of the three types of outliers. We reduce the state feature dimension to 2 via t-SNE and compute Mahalanobis distances of these two kinds of states to their centrality within each action class under the estimation based on MD or Robust MD, respectively. Fig. 9 suggests that Robust MD separates inliers and outliers better than MD on Breakout within a random action class, indicating its effectiveness in detecting RL evaluation. Similar results are also given in other games.

We plot the distributions of inliers and three types of outliers on SpaceInvaders and Asterix games in Figs. 10 and 11, respectively. It is worth noting that Robust MD is also capable of enlarging the separation of distributions between inliers and both random and adversarial outliers on SpaceInvaders game, while its benefit seems to be negligible on OOD outliers (Breakout) on SpaceInvaders games as well as in Asterix game. We speculate that it is determined by the game's difficulty. Specifically, the PPO algorithm can

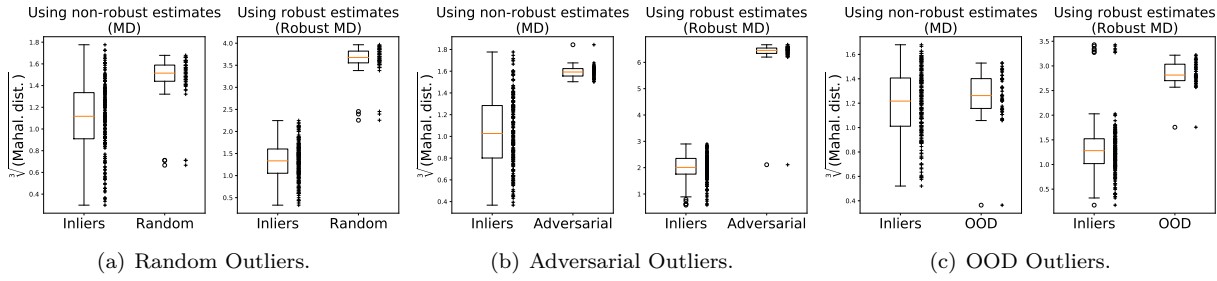

Figure 9: Boxplot of distributions between inliers and three types of outliers in an action class on Breakout game.

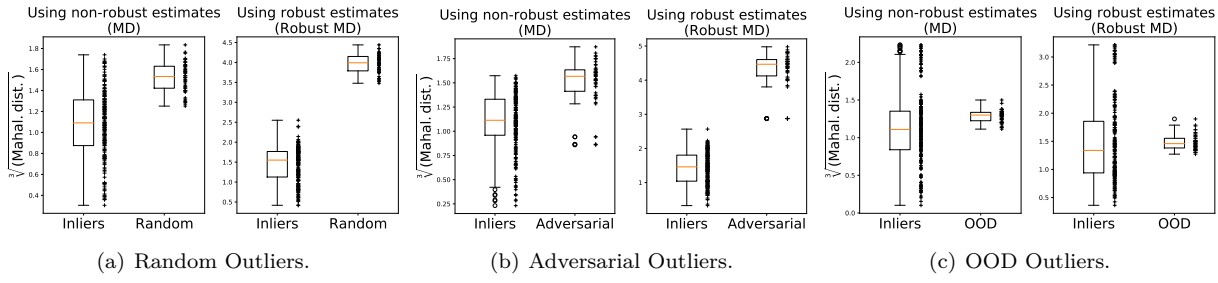

Figure 10: Boxplot of distributions between inliers and three types of outliers in an action class on SpaceInvaders game.

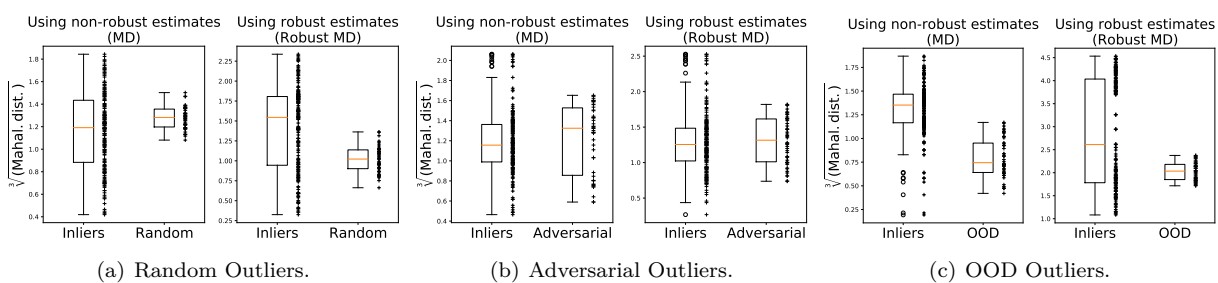

Figure 11: Boxplot of distributions between inliers and three types of outliers in an action class on Asterix game.

achieve desirable performance on the simple Breakout game, thus yielding informative feature space vectors. By contrast, there is room for the generalization of PPO on both SpaceInvaders and Asterix games, such that Robust MD might not help when handling the less meaningful state feature vectors in these two games.

## B.4 Sensitivity Analysis

We provide the sensitivity analysis of Robust MD in terms of the PCA dimension in Fig. 12. The impact of the number of principal components on the detection performance for robust MD detection is shown in Fig. 12. The detection accuracy over all considered outliers improves as the number of principal components increases, except for a slight decline for random and adversarial outliers (red and blue lines) on the Breakout game. The increase implies that the subspace spanned by principal components with small explained variance also contains valuable information for detecting anomalous states from in-distribution states, which coincides with the conclusion in (Kamoi & Kobayashi, 2020).

The result of MD estimation manifests in Fig. 13. It suggests that there is still an ascending tendency of detection accuracy as the number of principal components increases.

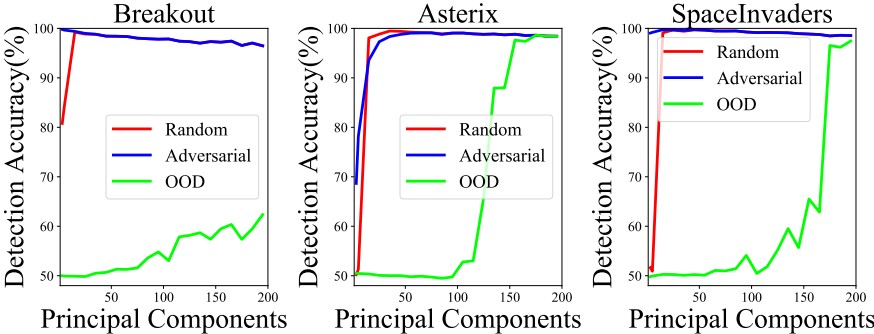

Figure 12: Detection performance under **Robust MD** as the number of principal components increases.

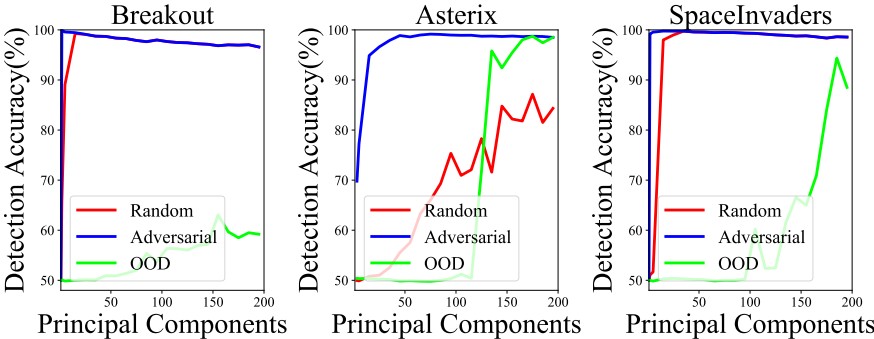

Figure 13: Detection performance under **MD** as the number of principal components increases.

## C    Results in Online Setting

### C.1    Setup and Full Main Results

As a supplement to the results on the main pages, we provide the whole results on two feature-input tasks and six Atari games from Fig. 14 to Fig. 21. The "Mean Score" in the first row indicates the accumulated rewards of PPO, and the "F1 Score" in the second row shows the detection performance during RL training. The F1 score is computed based on precision and recall. The third row shows the relationship between the average F1 score and policy performance during training. We can find that higher detection accuracy is generally associated with better policy performance. We also find that the cumulative reward is not strongly correlated with detection ability in some games. A high detection accuracy may only improve the cumulative reward to a small degree. This suggests that we need more metrics to measure the effect of our detection performance more effectively. Hyperparameters in our methods are shown in Table 7.

| Hyperparameter | Value |
| --- | --- |
| Confidence level $(1\text{-}\alpha)$ | 1-0.05 |
| Moving window size $(m)$ | 5120 |
| Sample size $(N_c)$ | 2560 |
| Iteration $(K)$ | $\approx 10000$ (1e7 steps in total) |
| Environment number $(N)$ | 8 |
| Horizon $(T)$ | 128 |

Table 7: Hyper-parameters in the training phase. RL-related parameters are the same as those of the PPO algorithm.

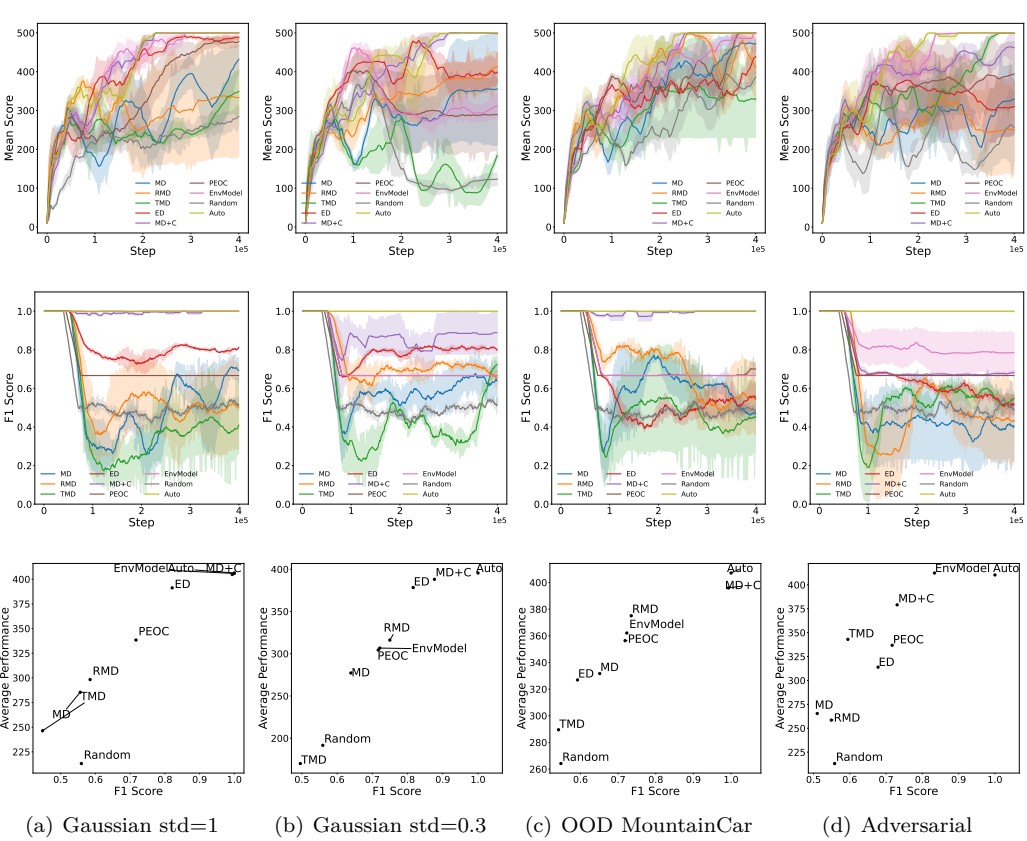

Figure 14: Detection performance across various state outliers in the online training on CartPole.

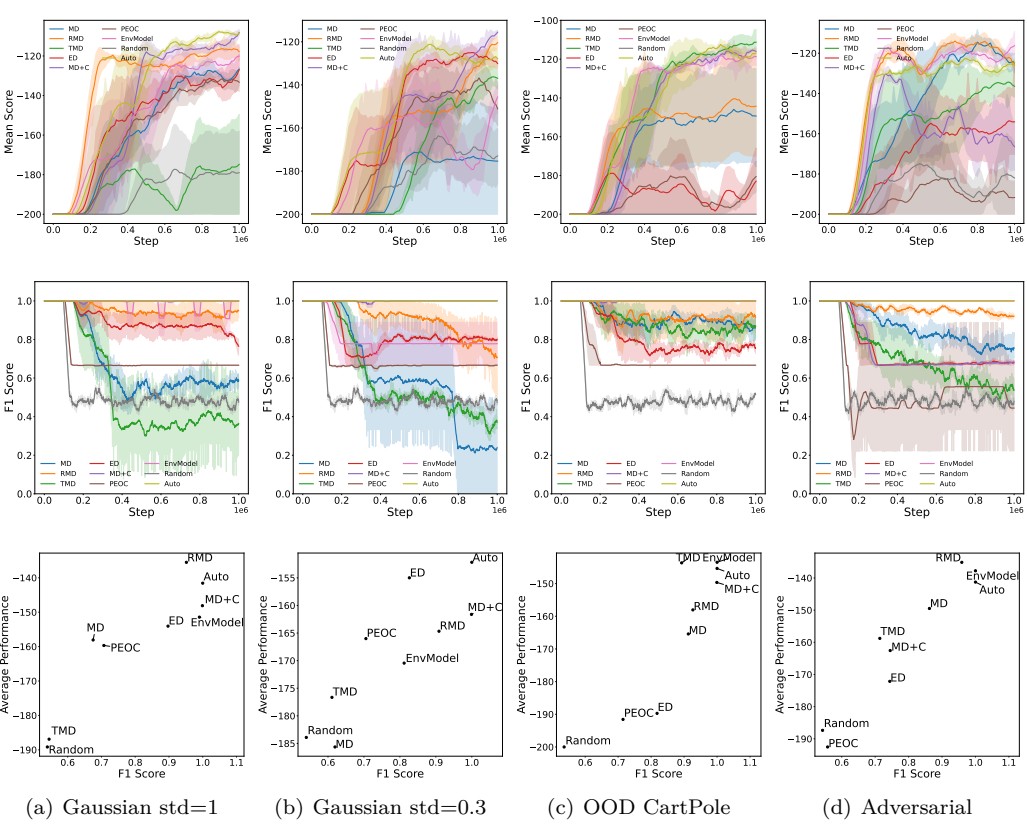

Figure 15: Detection performance across various state outliers in the online training on MountainCar.

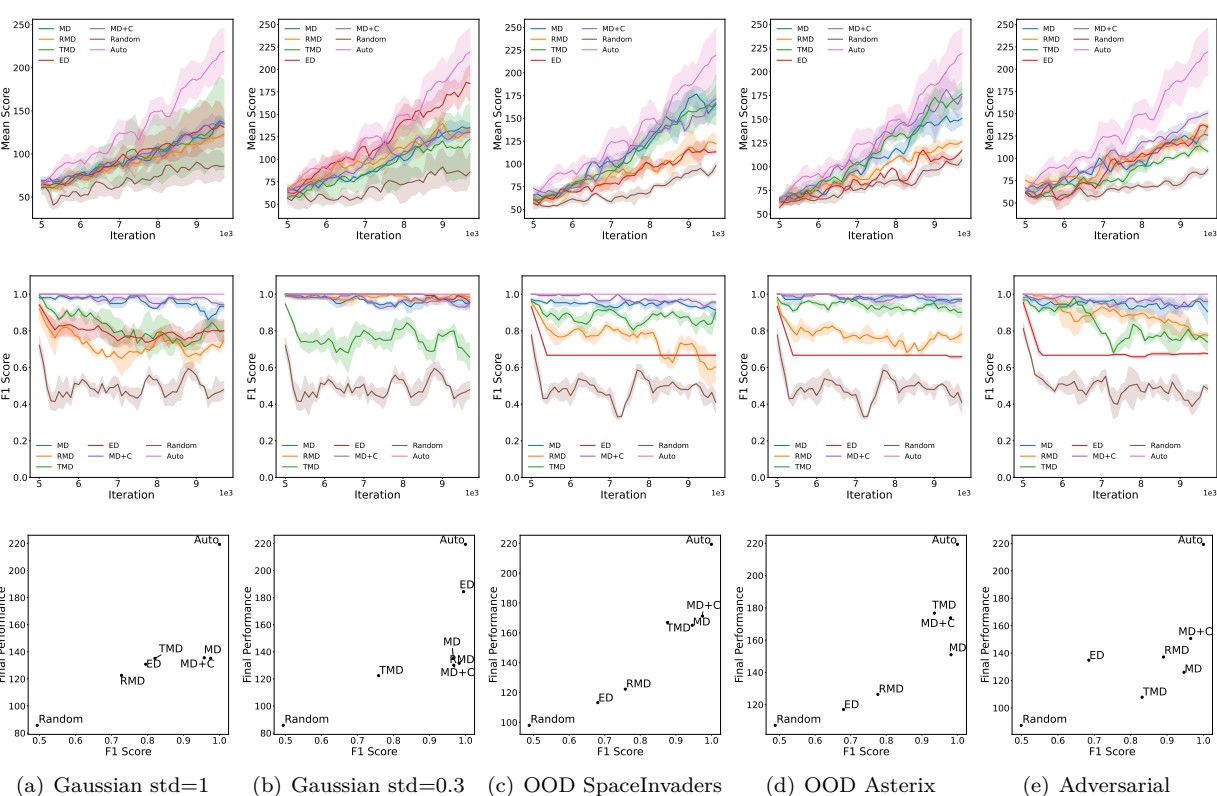

Figure 16: Detection performance across various state outliers in the online training on Breakout.

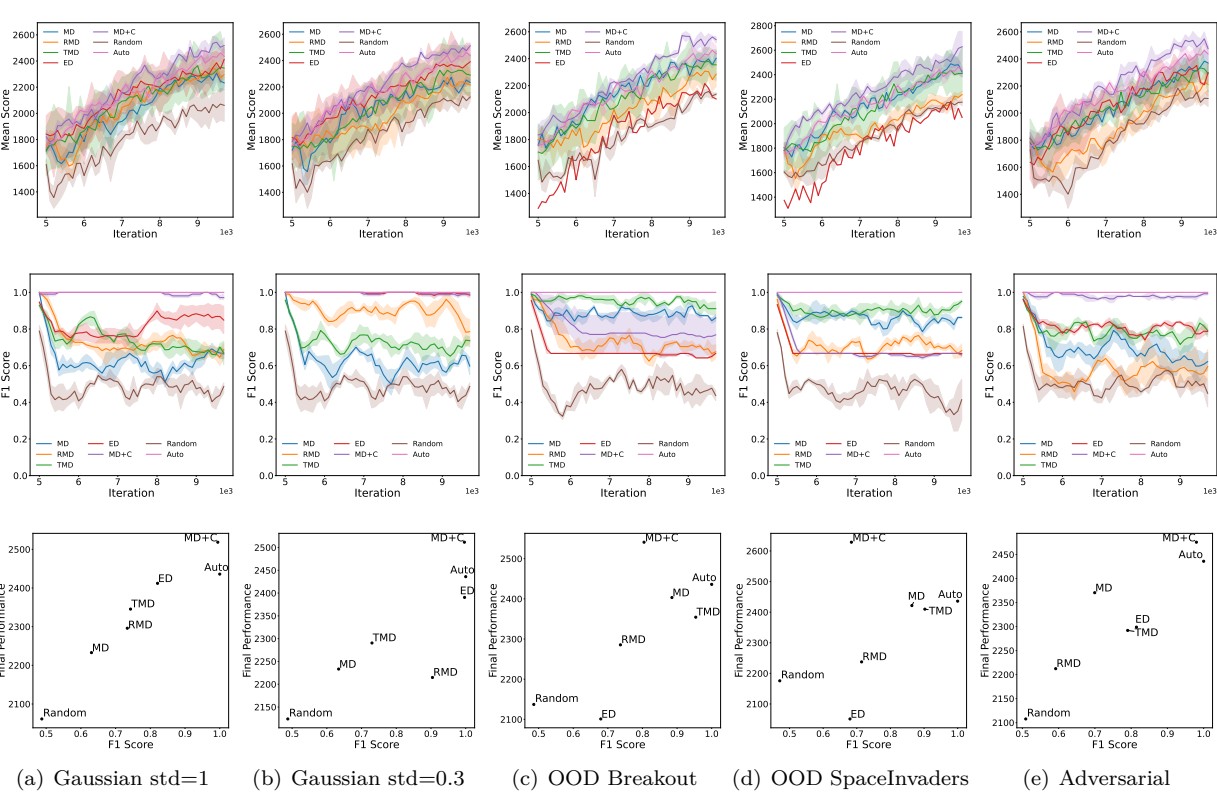

Figure 17: Detection performance across various state outliers in the online training on Asterix.

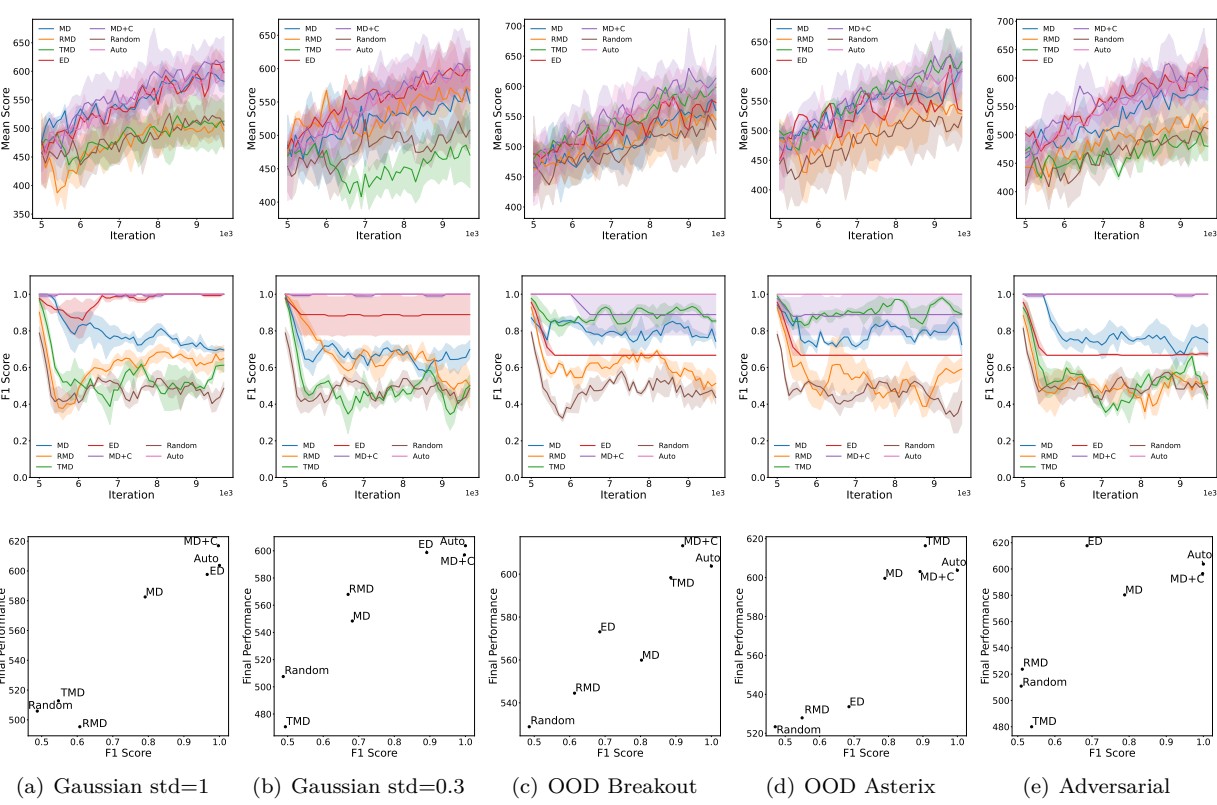

Figure 18: Detection performance across various state outliers in the online training on SpaceInvaders.

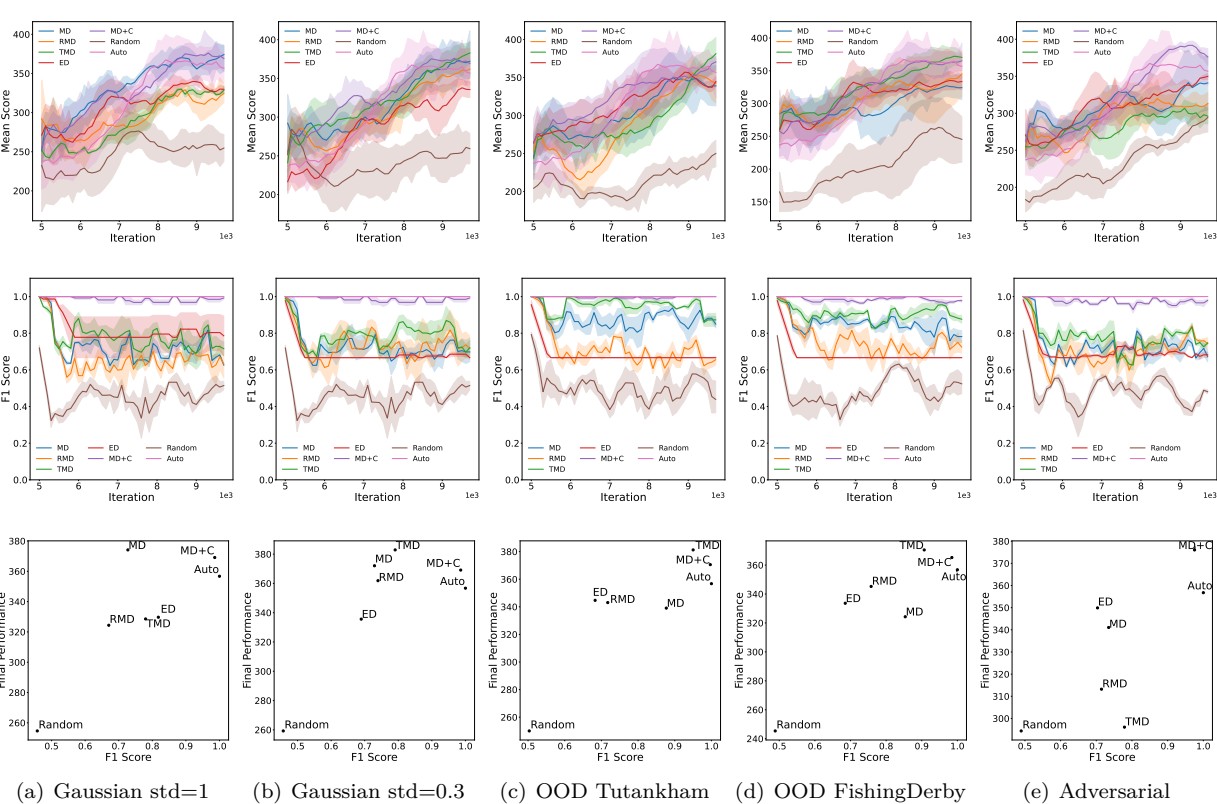

(a) Gaussian std=1    (b) Gaussian std=0.3    (c) OOD Tutankham    (d) OOD FishingDerby    (e) Adversarial

Figure 19: Detection performance across various state outliers in the online training on Enduro.

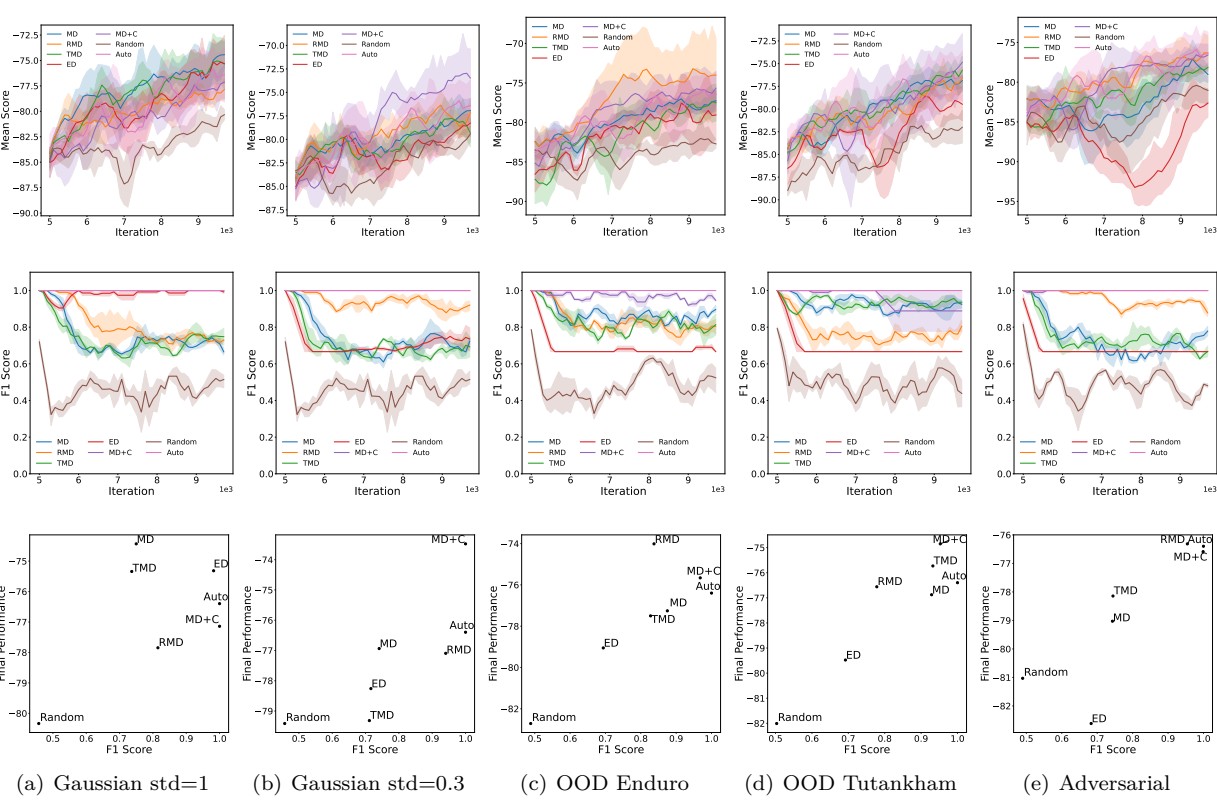

Figure 20: Detection performance across various state outliers in the online training on FishingDerby.

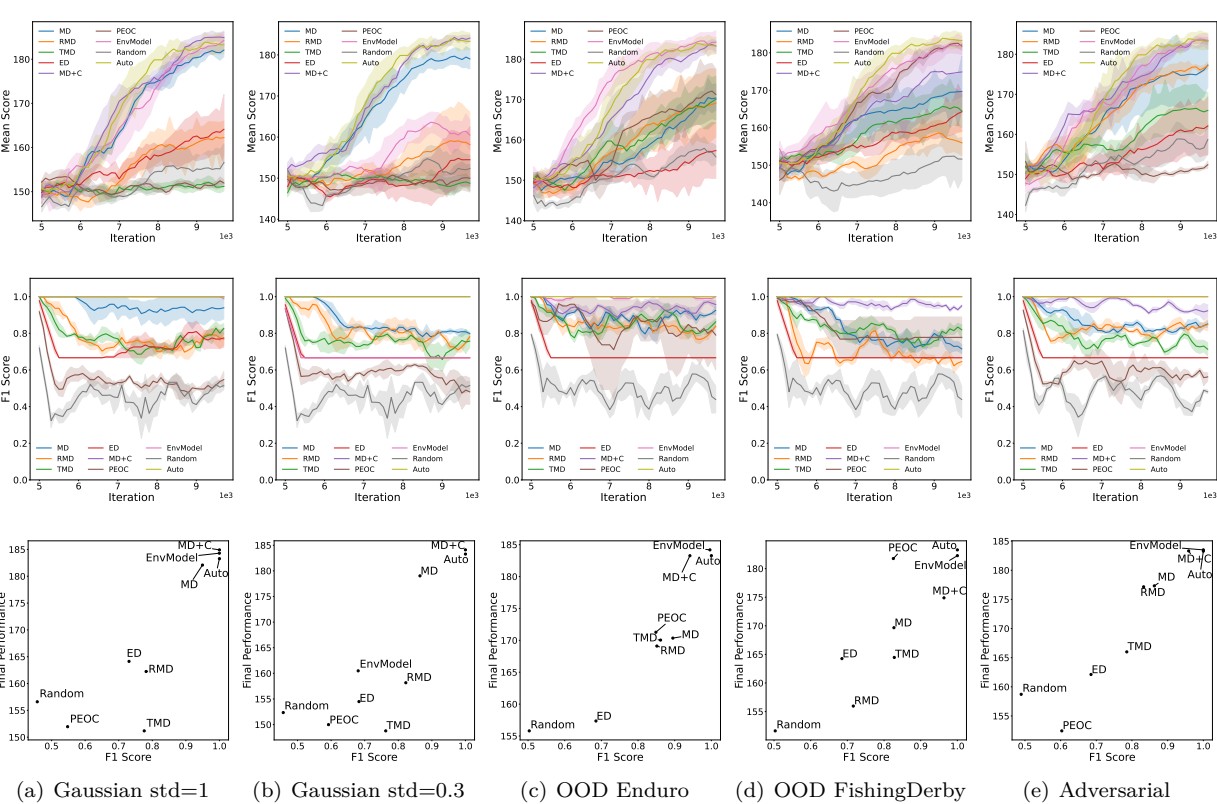

Figure 21: Detection performance across various state outliers in the online training on Tutankham.

## C.2    Ablation Study on Double Anomaly Detectors

Fig. 22 reveals that double self-supervised detectors can help adjust the detection errors and improve the detection accuracy compared with the single detector. MD with double detectors outperforms MD with a single detector significantly, although RMD with double detectors is comparable to RMD with a single detector.

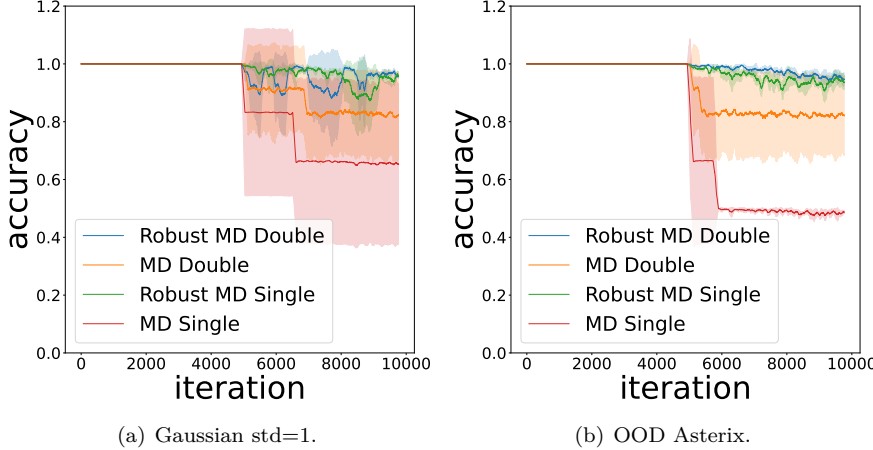

(a) Gaussian std=1.  (b) OOD Asterix.

Figure 22: The detection accuracy with and without double self-supervised detectors on Breakout with random and OOD outliers on Breakout.

## C.3    Ablation Study on Number of Noisy Environments

We train PPO in two, four, or six noisy environments with random and OOD outliers among all eight parallel environments. We use PCA to reduce the feature vectors to 50 dimensions and estimate the detector using Robust MD. Fig. 23 illustrates that compared with the **Auto** baseline, our RMD method is robust when encountering different ratios of outliers, especially with a higher contamination ratio. The dashed lines in different colors represent **Auto** baselines that correspond to the different number of noisy environments. The training performance with our detection method gradually approaches the ideal baselines, i.e., **Auto**.

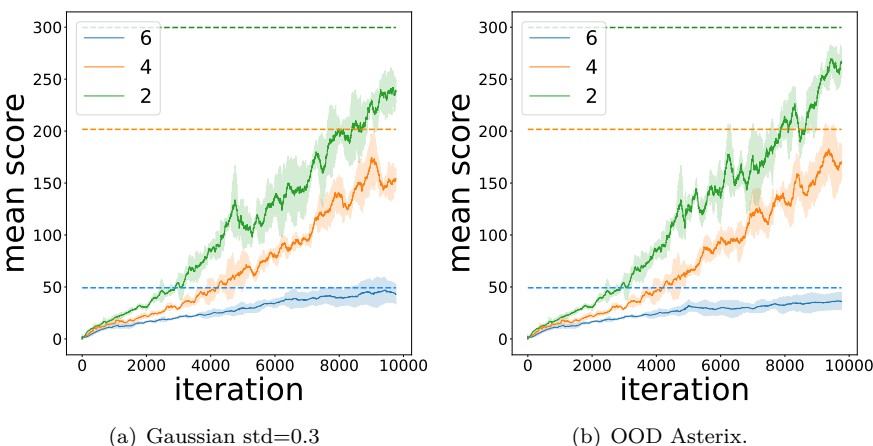

(a) Gaussian std=0.3.  (b) OOD Asterix.

Figure 23: Training performance under Robust MD detection under different proportions of outlier exposure on Breakout (2, 4, 6 out of 8 environments).

