# OpenReview forum: "A Distance-based Anomaly Detection Framework for Deep Reinforcement Learning"
_TMLR — Accepted by TMLR_

### Review · Reviewer_8DuC · 2024-08-21

**Summary Of Contributions:**

This paper is about a crucial issue of anomaly detection in deep reinforcement learning. It proposes methods to detect outliers in deep RL based on Mahalanobis Distance(MD), and improve it to robust MD-based detection and conformal MD-based detection. While many distance based outlier detection metrics rely on a Gaussian assumption to construct the detection threshold, this paper utilizes conformal prediction to get rid of the Gaussian assumption, which brings the theory closer to reality without many constraints. This paper considers the problem of anomaly detection in both offline and online RL settings. The paper also includes experiments in each setting to verify the effectiveness of the MD based outlier detection methods, within which the conformal MD method achieves the best detection performance in most scenarios.

**Audience:**

Yes

**Broader Impact Concerns:**

Not related.

**Claims And Evidence:**

Yes

**Requested Changes:**

Requested changes:
1. It will be more convincing to add other state of art outlier detection methods to compare. Outlier detection is a common problem involved in many transfer learning algorithms, like this paper “Bridging Theory and Algorithm for Domain Adaptation”. So for example, how the distance based metric is compared with neural network based discriminator to detect outliers?
2. Now most of the outliers/anomalies are artificially made, like adding Gaussian noises and so on. How would this outlier detection method perform with some real domain shift datasets?
3. It would be helpful to declare how the outlier detection is related to policy performance improvement.

**Strengths And Weaknesses:**

Strengths:
1. This paper does thorough analysis of the MD based anomaly detection methods for both online and offline experiments. And the writing is clear and easy to understand.
2. The introduction of conformal prediction with distance based outlier detection reduces the assumption of Gaussian distribution, where in many cases Gaussian distribution assumption is actually false. Also as expected, the conformal MD method achieves the best result.

Weaknesses:
1. In the experiments, it only compares with other distance based methods and among the methods proposed in this paper. But lack experiments about how these distance based methods would compare with other types of state-of-art outlier detection methods.
2. It is not written clearly how the outlier detection is related to the policy performance improvement. I see in the online RL experiments part, some results about rewards are shown. But not sure how the paper wants to show the correlations.

---

> ### Author Response · Authors · 2024-09-14
>
> > 1. It will be more convincing to add other state of art outlier detection methods to compare. Outlier detection is a common problem involved in many transfer learning algorithms, like this paper “Bridging Theory and Algorithm for Domain Adaptation”. So for example, how the distance based metric is compared with neural network based discriminator to detect outliers?
>
> Thanks for this suggestion. We have read this paper carefully, but we find that the method is particularly designed for domain adaptation, which requires the target data distribution. In contrast, our method focuses on anomaly detection in the current task, identifying the domain boundaries associated with a well-trained or constantly updated policy. To provide a more comprehensive comparison, we added two related baseline methods: PECO [1], which uses the policy entropy as the detection metric, and EnvModel [2], which additionally trains a neural network based probabilistic dynamic environment model and employs ensemble techniques. Please refer to the revised ``Baseline Methods`` in the experimental section and Tables 1 - 4 and Figure 4 - 5 for the new empirical results. It suggests that the new baselines are less effective than our proposed distance-based detection methods.
>
> [1] Sedlmeier et al. Policy entropy for out-of-distribution classification. ICANN 2020.
>
> [2] Haider et al. Out-of-distribution detection for reinforcement learning agents with probabilistic dynamics models. AAMAS 2023.
>
> > 2. Now most of the outliers/anomalies are artificially made, like adding Gaussian noises and so on. How would this outlier detection method perform with some real domain shift datasets?
>
> Firstly, Gaussian and adversarial noises are artificially crafted to mimic measurement errors and adversarial perturbations, respectively. However, OOD outliers are generated from other environment simulators, largely representing the real domain shift datasets.
>
> Moreover, as online RL requires interaction with the environment to collect data, it is challenging to directly apply our method to the real world. However, simulators serve as ideal testbeds for RL problems. We also searched for real-world datasets containing (state, action, reward, next state) tuples for offline settings, but the most commonly used datasets are also simulated [1]. This is why we further tested our method on CARLA, an open-source simulator for autonomous driving research known for its high-quality rendering and realistic physics, to demonstrate the effectiveness of our method in a more realistic setting.
>
> [1] Fu et al. D4rl: Datasets for deep data-driven reinforcement learning. 2020.
>
> > 3. It would be helpful to declare how the outlier detection is related to policy performance improvement.
>
> Thanks for this great suggestion. We have added scatter plots between the detection accuracy and policy improvement across different kinds of state outliers in Figures 4 and 5 of Section 7.2 as well as figures in Appendix C.1. These scatter plots demonstrate that a more accurate detector generally implies a larger policy improvement.

---

### Review · Reviewer_e3pr · 2024-09-02

**Summary Of Contributions:**

This paper introduces an anomaly detection framework based on Mahalanobis Distance based on hypothesis testing for use in Deep RL settings. The proposed anomaly detection framework functions in both offline and online learning settings by virtue of adapting the proposed framework via using moving window estimation and double self-supervised detectors in online settings, where new data is collected through time.

**Audience:**

Yes

**Broader Impact Concerns:**

This paper introduces a preliminary proof of concept about anomaly detection within RL settings. It does not propose a solution to real-world use cases and should not be used as such. I do not have any broader impact concerns in this regard since the methodology and proposed framework has not been rigorously evaluated in sim-to-real settings and I strongly doubt that anyone who was working in these settings would employ the proposed framework without additional safeguards in place.

**Claims And Evidence:**

Yes

**Requested Changes:**

From the weaknesses and other comments above:
 - Provide a more detailed description of the intended RL problem setting that MDX (and/or all distance-based anomaly detection methods)  is situated.
 - Revise the narrative structure to be either 100% for introducing and testing a new framework or 100% as a study/investigation into distance based anomaly detection.
 - Better framing of anomaly detection vs. RL generalization and/or meta RL considerations of OOD-ness.
 - Discuss how MDX may fit into an offline-to-online learning environment.
 - Provide more information about how the policy is pre-trained in online experiments. What is meant by pretraining? What is the specific set-up for this?

Additional changes:
It would be helpful if summary sentences were included at the end of each paragraph of Section 2, directing the reader to understand how the content relates to the proposed solutions and how the prior work lays the foundation for these advances. The final paragraph “Conformal Prediction and Conformal Anomaly Detection” provides a great example of where this works well.

**Strengths And Weaknesses:**

This paper formally studies anomaly detection in RL. I’m not overly convinced by the authors’ claims that there has not been extensive study in these directions. It may just be a matter of semantics, but I believe that there is substantial overlap with considerations within MetaRL and generalization studies that need to be more fully considered as components of anomaly detection in this work. One such paper is Ajay, et al (2022 NeurIPS) that addresses detecting and adjusting for meta RL agents drifting out of distribution in subsequent tasks.

While I acknowledge that there are differences between anomaly detection and the above from a fundamental perspective, I do however believe that differentiating or more clearly describing what is meant by the authors regarding generalization vs. anomaly detection. If this paper is solely focused on detection and not mitigating erroneous behaviors when the agent drifts OOD, then that’s fine. I think that we should be more clear here where the paper is directed and what it’s limitations are.

I enjoyed this paper. I am not entirely sure that distance based measures are the appropriate form of determining anomalies but it does appear to be serviceable based on the experimental results. I appreciated that the authors took time to develop multiple approaches for detecting anomalies within the Mahalanobis distance formulation and considered them all equally. I appreciated the extent with which the experiments considered each approach (and appropriate simple baselines) equally. I am however slightly disappointed that there were not additional anomaly detection baselines included in the results from the methods introduced in Section 2. There was a claim that distance based methods are better suited for the envisioned task but it is difficult to gauge whether this claim is satisfied by virtue of the included experiments.

However, I found the development of the proposed MDX method to be easy to follow and relatively straightforward. To this end, I think that the paper does a good job introducing the primary contributions even though the communication of them may have been a bit muddled (see the various weaknesses and questions below).

There are several weaknesses that I noted throughout the paper and I enumerate them below via questions that I seek clarity on.

Does MDX assume that there are a discrete, finite set of actions? This is an important assumption/limitation that should be mentioned up front. As noted in the footnote on page 5, this can be extended to continuous action spaces but there are possibly computational overheads as well as tradeoffs with the resulting binning strategy to discretize the action space in such settings? Overall, there could be a much clearer description of the types of problems that MDX is envisioned to solve.

One major weakness of the proposed framework is that there is fairly little maturity about what the intent of an anomaly detector is. How does this change the learning/operating environment of the RL algorithm? There was some slight nod toward this in the introduction but I found that there is some aspect of further discussion and/or development of this idea and where it fits into an intended use case to be lacking.

Not necessarily a weakness but a consideration of potential offline-to-online settings is missing. In this vein, I’m surprised that the authors did not connect their proposed framing within the risk-sensitive RL literature. There is a significant amount of overlap with the terminology and motivations (“unsafe actions”, etc) that exists there.

I think that it needs to be noted more formally that there are distinct approaches under the framework when operating in either offline or online settings and the choice of one form of MDX warrants a different setup, computational burden, etc. This is kind of difficult to wrap my head around. The paper seems to be proposing a single solution (MDX) but then at the end transitions the narrative to be a survey or study of distance based anomaly detection approaches in RL. I think that the paper would be greatly improved if the authors decided on a single narrative. I think that the strongest one would be that the paper is a survey of distance based anomaly detection approaches in RL (both offline and online) rather than attempting to proposed a single method/framework as a "novel" contribution.

I appreciated the differentiation made between prior anomaly detection approaches within RL to the proposed distance-based approach. However, how reliable is a distance measure in highly complex, high dimensional state-space domains? This is mitigated somewhat by using dimensionality reduction approaches over the feature space, but I think that this is a consideration about the appropriateness of the proposed anomaly detection framework.

The paper perhaps could be better structured by introducing a more general overview of how MDX is constructed, building from the foundations of the prior literature and background (Sections 2 and 3), before specifying specific variations on top of the framework for Offline (current Section 4) and Online RL (current Section 5). This is, of course, if the authors decide to maintain their current intended framing of introducing a singular framework (which I do not believe the paper supports at this time). The current structure, with some rewriting, could be maintained if the paper was changed to more appropriate be a study into the effectiveness of different types of distance-based anomaly detection approaches.

I’m not entirely certain that all of Section 4.1 is necessary to motivate and introduce what is currently in Section 4.2. They could each be condensed and combined into a single section. Since there is a clear tonal shift in Section 4.3, I think that the writing could be improved by more directly getting to the conformal, distribution free detection approaches used at the core of what may be the proposed MDX framework (if the narrative structure of the paper is to focus on a singular framework). The optionality of switching between the Chi-Squared thresholding (Gaussian assumption) vs. the conformal approach is not totally clear. I believe that the paper would be improved if one approach is specified and the other is used as a potential ablation to justify the other? Currently, the beginning of Section 4.4 is confusing due to the optionality of nominal MD vs. robust MD, Gaussian assumption vs. conformal detection, etc.

In online settings, are there steps to re-identify if inliers or outliers may have changed based on the shifting of the policy distribution? I wonder if there is not additional robustness that may be found through the reinvestigation of whether Inlier or Outlier status may have changed with the policy. Of course, this may not be necessary but it could be worth re-visiting? This may not be a relevant question if the moving window casts away data that is "too old".

The moving window estimation section is missing sufficient technical description of what is happening. Is the moving window corresponding with the iteration/time of the policy updates? What happens when the data falls outside that window? Is it cast away? Is there a concern about catastrophic forgetting when the anomaly detection mechanism is only seeded with the most recent data? This is perhaps of a lesser concern when operating in episodic settings with restarts, where the initial state distribution may remain fixed, but this has not been specified clearly by the authors.

It's unclear what is meant toward the end of Section 5 (regarding Online RL) about pre-training a policy before performing anomaly detection procedures. I think that this is a fairly important detail about how the MDX framework is expected to be evaluated and used that is omitted from this paper. From this, I am fairly uncertain what takeaways I have about the work.

In Appendix B.4, it is shown that an increase in the number of retained features aids in anomaly detection (not surprising given the high dimensionality of the provided image observations). However, the choice of 50 features is not explained for the main results provided in the paper. Why? Especially since 50 appears to be the lower end of what was tested in the experimental analysis.

In the CARLA experiments, why are there no learning curve analyses? If this is to be included in the online RL section, and with the prior emphasis on the MDX framework’s adaptations for this learning setting, it feels a bit incomplete to not have some indication of how the learning progresses through time and with the introduction of anomalous observations.

Corresponding references:

#### Ajay, Anurag, et al. "Distributionally adaptive meta reinforcement learning." Advances in Neural Information Processing Systems 35 (2022): 25856-25869.

---

> ### Author Response · Authors · 2024-09-14
>
> > 1. This paper formally studies anomaly detection in RL. I’m not overly convinced by the authors’ claims that there has not been extensive study ...
>
> While the distributional shift in RL context is related to many fields, including multi-task, transfer, continual, meta RL, and generalization, each of these areas emphasizes different aspects. For instance, meta RL, in Ajay, et al (2022 NeurIPS) learns meta-policies by widening uncertainty sets at training time, thus potentially adapting to a variety of test-time distribution shifts. By contrast, anomaly detection aims to cast an alert in a decision-making system whenever they encounter unfamiliar observations, promoting a reliable generalization/training process in the current environment instead of other test tasks.
>
> > 2. While I acknowledge that there are differences between anomaly detection and the above from a fundamental perspective, I do however believe that differentiating or more clearly describing what is meant by the authors regarding generalization vs. anomaly detection ...
>
> This paper is indeed solely focused on outlier detection instead of generalization on next tasks and thus clearly differentiates meta, transfer, and robust generalization in RL. To emphasize this discrepancy, we have added one paragraph in the related work of the revised version to discuss the distribution shift issue in RL.
>
> > 3. I enjoyed this paper... I am however slightly disappointed that there were not additional anomaly detection baselines included in the results from the methods introduced in Section 2...
>
> Thanks for recognizing our rigorous evaluation of our proposed outlier detection approaches. We have added two new baselines beyond the distance-based detection framework, including the entropy based method (PECO) [1] and a model-based detection (we name it EnvModel) [2]. Please refer to Tables 1-4 and Figures 4-5 for detailed results. The new baselines are less effective in detecting various kinds of outliers than our distance-based methods.
>
> [1] Sedlmeier et al. Policy entropy for out-of-distribution classification. ICANN 2020.
>
> [2] Haider et al. Out-of-distribution detection for reinforcement learning agents with probabilistic dynamics models. AAMAS 2023.
>
> > 4. Does MDX assume that there are a discrete, finite set of actions ...
>
> Our MDX detection framework focuses on the discrete action class and can be extended to the continuous action regime by discretizing the action into multiple bins, as mentioned in the footnote. To have a clearer description of our detection problem, we have revised and further highlighted the setting our MDX focuses on in the footnote on Page 6.
>
> > 5. One major weakness of the proposed framework is that there is fairly little maturity about what the intent of an anomaly detector is ...
>
> We highlight the motivation for anomaly detection in the paragraphs ``practical Scenarios`` and ``Motivating Examples`` and in Figure 1 in the introduction section. In the offline setting with a fixed policy, state outliers tend to cause unsafe behaviors for RL systems, such as the case of autonomous driving in Figure 1(a) and the performance degradation in Figure 1(c). Acting on anomalous data could result in hazardous situations, and thus, developing suitable anomaly detectors for RL is particularly important in safety-critical scenarios. We have also added this discussion in Section 4 and in the introduction of offline and online settings on Page 2 of the revised paper.
>
> > 6. Not necessarily a weakness but a consideration of potential offline-to-online settings is missing. In this vein, I’m surprised that the authors did not connect their proposed framing within the risk-sensitive RL literature ...
>
> Our work focuses on offline and online settings separately, while leaving the extension between their interaction, e.g., offline-to-online setting, as the future work. The risk in risk-sensitive RL typically refers to the uncertainty of the potential outcomes [1], involving risk-averse and risk-seeking policies in exploring environmental uncertainty. Instead of risk-sensitive RL, we believe the reviewer is referring to safe RL, which often considers distributional shifts and adversarial robustness. Our work falls into the realm of safe RL, particularly focusing on anomaly detection.
>
> [1] Will Dabney et al. Implicit Quantile Networks for Distributional Reinforcement Learning. ICML 2018.

---

> > ### Comment · Reviewer_e3pr · 2024-09-15
> > **Re: "safe RL"**
> >
> > Thank you for clarifying this. I have a better sense of where the authors intend to position their paper. As such, I believe that a sentence or two, even a short paragraph discussing this intention should be added to the Introduction. This would greatly improve the foundations of the work as well as help connect to the relevant operating literature within the safe RL community.

---

> > > ### Author Response · Authors · 2024-09-15
> > > **Author Further Response: safe RL**
> > >
> > > We are pleased that our response has helped clarify our contribution to you. In line with your suggestions, we have further emphasized the safe RL setting in the first paragraph of the Introduction. We have also added more discussion in the middle of Page 2 to further elaborate on this point, particularly connecting relevant literature in the last paragraph of the Related Work section. These revisions more appropriately position the contribution of our study. We hope the updated version of our paper meets your expectations, but if there is anything else that requires clarification or further discussion, please do not hesitate to let us know.

---

> ### Author Response · Authors · 2024-09-14
>
> > 7. I think that it needs to be noted more formally that there are distinct approaches under the framework when operating in either offline or online settings ...
>
> Our study is to propose a single MDX detection framework instead of being viewed as a survey of detection methods. Within the MDX detection framework, we generally introduce two variants, including distribution-based and distribution-free detection. Distribution-based detection relies on the Gaussian assumption, and the detection is applied based on the asymptotic Chi-squared distribution. By contrast, distribution-free detection leverages conformal inference to calibrate the detection threshold. However, all the variants are well organized within the single distance-based MDX framework we proposed in this study.
>
> > 8. I appreciated the differentiation made between prior anomaly detection approaches within RL to the proposed distance-based approach. However, how reliable is a distance measure in highly complex, high dimensional state-space domains ...
>
> We argue that constructing detection in the low-dimensional feature space (by dimension reduction) is a natural and computationally effective strategy to tackle the highly complex and high-dimensional state-space domain. While other detection approaches are also possible in RL beyond the distance measure, the distance-based method is popular and often viewed as the first choice in developing anomaly detection. Also, we empirically substantiate the effectiveness of our proposed distance-based detection strategy across various kinds of adopted environments, compared with other baselines beyond the distance measure.
>
> > 9. The paper perhaps could be better structured by introducing a more general overview of how MDX is constructed ...
>
> Thanks for this great suggestion. As we aim to structure our paper by maintaining a distance-based detection framework in RL, we thus add one section to introduce the general framework of MDX before offline and online settings in the revised paper. Please refer to the new section 4 in the revised paper for more details.
>
> > 10. I’m not entirely certain that all of Section 4.1 is necessary to motivate and introduce what is currently in Section 4.2 ...
>
> Following your suggestion, we combine vanilla MD-based detection in Section 4.1 and the robust variant in Section 4.2 in a single section (new Section 5.1 in the revised paper), as both rely on the Gaussian assumption. As such, our MDX framework generally encompasses two classes of detection methods in RL, i.e., the distribution-based detection and the distribution-free/conformal one. We highlight that switching from the Chi-Squared/distribution-based approaches to the distribution-free/conformal is well-motivated due to the potential violation of Gaussian assumption and the resulting ineffective detection. The organization in Section 4.4 in the original paper is to integrate all detection methods within one single algorithm framework, where robust MD and conformal MD are seen as the variants of the original MD approach.
>
> > 11. In online settings, are there steps to re-identify if inliers or outliers may have changed based on the shifting of the policy distribution ...
>
> In typical RL settings, the agent takes actions given the states that come sequentially without intentionally storing all past data. As you mentioned, the moving window estimate indeed casts away data that is ``too old``, and therefore, we do not think it is necessary for a re-visiting.
>
> > 12. The moving window estimation section is missing sufficient technical description of what is happening ...
>
> The moving window mechanism operates similarly to a first-in-first-out buffer, storing the most recent samples and discarding the oldest ones. The window size balances past and recent data used for detection. Data falling outside the window is cast away. As you mentioned, the episodic setting with restarts ensures there is no concern about catastrophic forgetting. To mitigate your concern, we have added necessary discussions in the revised paper.

---

> > ### Author Response · Authors · 2024-09-15
> >
> > > 13. It's unclear what is meant toward the end of Section 5 (regarding Online RL) about pre-training a policy before performing anomaly detection procedures ...
> >
> > As our detection method relies on features extracted from the penultimate layer of the policy network, we need to pre-train the policy to ensure that these features capture meaningful information about the environment. A randomly initialized policy may not provide useful features for detection. Moreover, from a practical perspective, deploying a randomly initialized policy in a real-world scenario is unreliable. Instead, it is common practice to use a pre-trained policy as a warm start and then further improve it, such as the recommendation system example we mentioned at the end of Section 6.
> >
> > > 14. In Appendix B.4, it is shown that an increase in the number of retained features aids in anomaly detection (not surprising given the high dimensionality of the provided image observations). However, the choice of 50 features is not explained for the main results ...
> >
> > The choice of a 50-dimensional feature instead of higher dimensions is for computational convenience, especially in the online setting, where robust MD is computationally expensive. To align with the online setting, we also reduce the state feature space to 50-dimensional via the dimension reduction in the offline setting.
> >
> > > 15. In the CARLA experiments, why are there no learning curve analyses ...
> >
> > Firstly, anomaly detection in autonomous driving is more practical and meaningful in the offline setting, as illustrated in the motivating examples and a practical scenario in Figure 1 (a). Additionally, since CARLA is a more complex environment with high-quality rendering and realistic physics compared to Atari games, training autonomous driving agents in the CARLA environment for all baselines is costly and beyond our computational capacity. Therefore, we focus on the offline setting in our empirical study.
> >
> > > 16. Additional changes: It would be helpful if summary sentences were included at the end of each paragraph of Section 2 ...
> >
> > Thanks for this great suggestion. We have further improved these summary sentences. Please refer to each paragraph in the related work section of the revised paper.

---

> > > ### Comment · Reviewer_e3pr · 2024-09-15
> > > **Re: CARLA experiments**
> > >
> > > Thanks for your responses. My confusion about the structure of your results section is that your CARLA experiments follow directly after those investigated for Online RL. I can understand that the more complex/"bigger" experimental results would be very desirable as a final statement about the proposed MDX approach. This confusion could be addressed in a couple of ways. First, place these experiments after (or within) Section 7.1 or clearly describe that this is an offline experiment with previously collected data. A quick scan of 7.3 doesn't really help introduce this.

---

> > > > ### Author Response · Authors · 2024-09-15
> > > > **Author Further Response: CARLA experiments**
> > > >
> > > > We appreciate you pointing out this clarity issue. Based on your suggestion, we have included a description of the offline setting in the first paragraph of Section 7.3, which hopefully could address your concern. Please review the further revised paper. Thank you very much for your consistent dedication to reviewing our work.

---

### Review · Reviewer_7cuR · 2024-09-05

**Summary Of Contributions:**

The paper "A Distance-based Anomaly Detection Framework for Deep Reinforcement Learning" presents a novel framework called MDX for anomaly detection in deep reinforcement learning (RL) systems. Its main technical contributions include:

**Mahalanobis Distance-based Detection**: The framework uses the Mahalanobis Distance (MD) within a hypothesis testing framework under Gaussian assumptions to identify random, adversarial, and out-of-distribution (OOD) state outliers. The MD-based detection is extended with a robust MD version using Minimum Covariance Determinant (MCD) to handle noisy data more effectively. The authors introduce a non-parametric conformal prediction method to address limitations of the Gaussian assumption, which allows for a distribution-free anomaly detection.

**Offline and Online Settings**: The framework supports both offline (fixed datasets) and online (dynamic datasets with continuous interaction) RL settings, making it flexible for real-world applications. In the online RL setting, the paper introduces a technique that maintains two self-supervised detectors—one for inliers and one for outliers—to improve detection performance as the policy evolves during training.

**Comprehensive Evaluation**: Extensive experiments are conducted in both Atari games and autonomous driving scenarios to demonstrate the effectiveness of the MDX framework in detecting multiple types of anomalies, thus improving the safety and reliability of RL systems.

**Audience:**

Yes

**Broader Impact Concerns:**

No ethical / broader impact concerns.

**Claims And Evidence:**

Yes

**Requested Changes:**

1. I know the work focuses on deep RL, but some experiments on standard RL environments, like openAI gym, should be reported.
2. It's better to add some downstream tasks for the offline setting.
3. Please revise the notation and writings, for example, in the second line of section 4.1, one $s_t$ is bold and another one is not. I assume they are the same $s_t$.

**Strengths And Weaknesses:**

Strength:
1. The writing is clear to follow and understand.
2. The authors perform comprehensive evaluations.
3. The empirical performance looks good.
4. The motivation parts are substantial and of high quality.

Weakness:
1. Discretizing policy to several classes might not be practical in some cases and suffers from the curse of dimensionality.
2. The computational cost is too heavy, maintaining the $\mu$ and $\Sigma$ for each class requires much more data.
3. The paper only evaluates image-input tasks, where there is a possibility that the shape of sampled distributions is nicer and more centered.
4. There is no guarantee that the method will work for those state-action pairs with low visit probability. There might not be enough samples to give a reasonable identifier.
5, As the authors addressed in the limitation, the algorithms might regard the new exploration activities as outliers.

---

> ### Author Response · Authors · 2024-09-14
>
> > 1. Discretizing policy to several classes might not be practical in some cases and suffers from the curse of dimensionality.
>
> Firstly, we acknowledge that our detection strategy focuses on the discrete action space, but discretizing policy into several classes is natural, especially in the continuous action space.
>
> Secondly, a desirable policy often takes only a small number of (deterministic) actions, even within the large action space. Our detection aims to ensure that the policy takes reliable actions, and thus we only need to maintain $\mu_c$ and $\Sigma_c$ for actions that are likely to be taken by the policy, and then detect states based on existing detectors. For example, in our online setting, we estimate the detector for an action class only when there are enough samples for that class ($N_c=2560$). When detecting, the Detection Mahalanobis Distance $M(s)$ for an incoming state $s$ is computed based on these existing detectors, which largely mitigates the curse of dimensionality.
>
> Lastly, while the computational cost will generally increase as the action class grows, the computation involved in the low-dimensional $\mu_c$ and $\Sigma_c$ can be efficient via dimension reduction. Moreover, other techniques, such as adaptive discretization and action space reduction, can be appropriately incorporated to address this issue in large action classes.
>
> > 2. The computational cost is too heavy, maintaining $\mu$ and $\Sigma$ for each class requires much more data.
>
> We highlight that our method only requires estimating the mean ($\mu$) and covariance ($\Sigma$) in developing detectors using maximum likelihood estimation. This computational cost is much smaller compared to training a neural network. Additionally, through dimension reduction, estimating the reduced low-dimensional $\mu$ and $\Sigma$ becomes efficient. In terms of data requirements, our method needs significantly fewer samples (approximately 5000 for each action that is likely to be taken by the policy) compared to off-policy methods like DQN, which maintain a replay buffer with a size of $10^6-10^7$.
>
> > 3. The paper only evaluates image-input tasks, where there is a possibility that the shape of sampled distributions is nicer and more centered.
>
> Our evaluation focuses on image input tasks because they are usually more complex than vector feature input in classical control environments. To address your concerns about generalizability, we additionally evaluate our proposed detection approaches on two feature-input classical control environments, MountainCar and CartPole, in Sections 7.1 (offline setting) and 7.2 (online setting) of the revised paper. Like image-input tasks, our detection methods are still effective in low-dimensional state spaces.
>
> > 4. There is no guarantee that the method will work for those state-action pairs with low visit probability. There might not be enough samples to give a reasonable identifier. 5, As the authors addressed in the limitation, the algorithms might regard the new exploration activities as outliers.
>
> We argue that we should focus on the real sample trajectory collected by either a well-trained or constantly updated policy instead of equally considering all state-action pairs in the whole sample space, which could even be computationally infeasible. A desirable policy often takes a small number of action classes, even among a large action space. Therefore, it is more reasonable to maintain detectors for actions that are likely to be taken by the policy, and state-action pairs with low visit probability will not have separate detectors. With regard to the difference between outliers and new exploration, we view it as an open problem in this field and leave it as our future work.
>
> > 5. Changes: I know the work focuses on deep RL, but some experiments on standard RL environments, like openAI gym, should be reported.
>
> We have added detection experiments on CarPole and MountainCar environments in Section 7.1 Table 1, Section 7.2 Figure 4, and more details in Appendix B and C, which suggest similar results and further demonstrate the effectiveness of our proposed detection approaches.
>
> > 6. Changes: It's better to add some downstream tasks for the offline setting.
>
> Thank you for the suggestion. While it is not clear what typical downstream tasks are necessary for further demonstration, we point out that our online setting can be viewed as a downstream task of the offline setting. In particular, we first pretrain a policy and then continue to train it online.  We would be happy to add an additional evaluation on other downstream tasks if necessary.
>
> > 7. Changes: Please revise the notation and writings, for example, in the second line of section 4.1, one is bold and another one is not. I assume they are the same $s_t$.
>
> Thanks for pointing out this typo. We have corrected and proofread the whole manuscript again. Should you come across any additional typos, please do not hesitate to let us know.

---

### Comment · Action_Editor_9vc6 · 2024-09-05
**Author response starts**

Dear authors,

All three reviews have been submitted. Would you please read the reviews and respond if you see necessary? You are also allowed to update the manuscript to incorporate the requested changes. The next two weeks will be a rolling discussion period with the reviewers. After that the reviewers will make their decision recommendations. Thank you!

Dear reviewers, thank you very much for your very valuable contribution!

Best,
AE

---

### Decision · Action_Editor_9vc6 · 2024-10-02

**Recommendation:** Accept as is

**Comment:**

This paper proposes anomaly detection methods for RL algorithms, which are able to cast an alert when an RL algorithm encounters an unfamiliar state. The core technical framework is based on Mahalanobis distance, and a number of different variants are investigated, spanning the settings of offline, online, Gaussian, and distribution-free. It is also shown that during RL training, using the proposed anomaly detector to filter the encountered states can increase the training robustness. Extensive experiments are conducted to demonstrate the effectiveness of the proposed methods.

The reviewers all agree that the paper presents valuable contributions to the field. The technical developments are solid, and the writing is clear. There were initially some concerns on the discretization of the action space and the associated computational cost, but these were cleared during the rebuttal. As suggested by the reviewers, more experiments were added in the revision.

A remaining concern after the rebuttal is that more baselines are suggested for the experiments, and this is essentially related to the clarity of the paper's scope. The paper studies two objectives, anomaly detection, and using the proposed detector to improve the training robustness. While the first objective is clear and self-contained, the second objective possesses some intuitive similarity to certain related topics, such as "risk-sensitive RL" or "safe RL", therefore the reviewers would appreciate more baselines of such types. However, a consensus is that this is good to have but not a must. The current form of the paper is of high quality, and meets the acceptance criterion.

Therefore the recommendation is to accept the paper as is. The authors are encouraged to further sharpening the paper's scope and revising the related exposition, particularly in the section of experiments, during the camera-ready revision.

**Audience:**

Yes

**Claims And Evidence:**

Yes